# ImProver: Agent-Based Automated Proof Optimization

**Riyaz Ahuja   Jeremy Avigad   Prasad Tetali   Sean Welleck**
Carnegie Mellon University
`{riyaza,avigad,ptetali,swelleck}@andrew.cmu.edu`

## Abstract

Large language models (LLMs) have been used to generate formal proofs of mathematical theorems in proofs assistants such as Lean. However, we often want to optimize a formal proof with respect to various criteria, depending on its downstream use. For example, we may want a proof to adhere to a certain style, be declaratively structured, or concise. Having suitably optimized proofs is also important for learning tasks, especially since human-written proofs may not optimal for that purpose. To this end, we study a new problem of automated proof optimization: rewriting a proof so that it is correct and optimizes for an arbitrary criterion, such as length or declarativity. As a first method for automated proof optimization, we present ImProver, a large-language-model agent that rewrites proofs to optimize arbitrary user-defined metrics in Lean. We find that naively applying LLMs to proof optimization falls short, and we incorporate various improvements into ImProver, such as the use of symbolic Lean context in a novel Chain-of-States technique, as well as error-correction and retrieval. We test ImProver on rewriting real-world undergraduate, competition, and research-level mathematics theorems, finding that ImProver is capable of rewriting proofs so that they are substantially shorter and more declarative in structure.

## 1 Introduction

The fundamental virtue of a mathematical proof is that it provides certainty: a deductive argument shows that the assumptions of a mathematical statement logically guarantee the conclusion. In practice, however, informal, natural-language proofs are prone to imprecision, ambiguity, and error. Using a formal language such as Lean (de Moura & Ullrich, 2021) removes such ambiguity and imprecision and enables a proof assistant to verify correctness down to the primitives of a formal axiomatic system.

Although any two correct formal proofs of a statement equally establish the validity of their conclusion, there are various criteria on which one of them may be preferred over another. When an expert formalizer finishes a proof, they always go back and revise it, aiming, for example, to improve readability and robustness. Instructors show their students how to shorten their proofs and structure them better, and the maintainers of Lean's Mathlib (mathlib Community, 2020) demand revisions to submissions to improve their robustness and adhere to style guidelines.

To this end, we study a new problem of *automated proof optimization*: rewriting a proof so that it is correct and optimizes a user-specified criterion such as length or readability. To mathematicians and formalizers, the ability to improve proofs automatically is invaluable to the maintenance and development of libraries for research and pedagogy alike. For example, the development of Mathlib as an evolving corpus maintained by hundreds of human formalizers requires strict guidelines to ensure efficient and generalized theorems — a task that proof optimizers can excel at automating, in order to generate proofs that rely on existing lemmas with concision and generalizability.

Moreover, automated proof optimization is not only useful in its own right, but also for the purposes of improving AI that can find proof on its own. At the very least, it provides a form of data augmentation: the limited amount of formal training data is currently a bottleneck for machine learning, and our methods provide ways of generating additional data automatically. More interestingly, our methods also provide a means of optimizing training data. For example, other work (Jiang et al., 2023)

suggests that a promising means for generating formal proofs is to have an LLM sketch a high-level outline of a proof that can be filled in by symbolic automated reasoning methods. For that purpose, it is useful to have a corpus of proofs that are written in such a structured form. Our methods provide means of generating such proofs from less structured ones.

Our work shows that naively applying LLMs to proof optimization falls short, often resulting in incorrect or poorly optimized proofs. We develop various improvements that can be applied on top of a black-box language model, including Chain-of-States prompting – an analogy to chain-of-thought prompting (Wei et al., 2022) that shows intermediate proof states, contextual information, error-correction, and retrieval. We incorporate these into ImProver: a large language model agent that rewrites proofs to optimize arbitrary user-defined metrics in Lean. We test ImProver on rewriting real-world undergraduate theorems, competition problems, and research-level mathematics, finding that ImProver is capable of rewriting proofs so that they are substantially shorter and more declarative in style.[1]

**Original (human-written)**

```
lemma lemma0 {α : Type} {p : α → α → Prop}
    (h1 : ∀ x, ∃! y, p x y)
    (h2 : ∀ x y, p x y ↔ p y x) :
    ∀ x, Classical.choose
        (h1 (Classical.choose (h1
    x).exists)).exists=x := by

-- PROOF START
intro x
obtain ⟨y, h1e, h1u⟩ := h1 x
have h2' : Classical.choose (h1 x).exists =
    y :=
    h1u _ (Classical.choose_spec (h1
    x).exists)
rw [h2']
obtain ⟨w, h1e', h1u'⟩ := h1 y
have h4 := Classical.choose_spec (h1
    y).exists
have hxw : x = w := by
    apply h1u'
    rw [h2]
    exact h1e
rw [hxw]
exact h1u' _ h4
```

ImProver **(length-optimized)**

```
lemma lemma0 {α : Type} {p : α → α → Prop}
    (h1 : ∀ x, ∃! y, p x y)
    (h2 : ∀ x y, p x y ↔ p y x) :
    ∀ x, Classical.choose
        (h1 (Classical.choose (h1
    x).exists)).exists=x := by

-- PROOF START
intro x
obtain ⟨y, h1e, h1u⟩ := h1 x
rw [h1u _ (Classical.choose_spec _)]
obtain ⟨w, h1e', h1u'⟩ := h1 y
rw [h1u' _ ((h2 _ _).mpr h1e)]
exact h1u' _ (Classical.choose_spec _)
```

Figure 1: ImProver automatically rewrites formal proofs to optimize a criterion such as length or readability while remaining correct. In this example, ImProver optimizes a human-written lemma from the 2022 International Math Olympiad (Question 2, solution from Compfiles (David Renshaw, 2024)) for length. ImProver's optimized proof is correct and more concise.

## 2 RELATED WORK

Recently there has been wide interest in automating theorem proving in interactive proof assistants; see (Lu et al., 2023; Li et al., 2024) for surveys. Indeed, at a high level, proof assistants constitute a sound verifier in a prover-verifier game (Anil et al., 2021), suggesting that a machine-learning based prover that interfaces with such a verifier is a natural next step for formal reasoning systems.

A typical approach to developing machine learning provers (Polu & Sutskever, 2020) is to train on a large corpus of mathematical proofs such as Lean's Mathlib (mathlib Community, 2020; Han et al., 2022; Polu et al., 2022b; Lample et al., 2022; Yang et al., 2023; Hu et al., 2024). A model learns from the distribution of proofs in the corpus, such as Mathlib-style proofs. Recently, the AlphaProof (AlphaProof & Teams, 2024) system was shown to produce proofs with an arcane, non-human structure and syntax. We consider the new problem of rewriting a proof to optimize a metric, such as rewriting a proof into a more declarative or more concise one. Proof optimization is more general than theorem proving, since we can also rewrite an empty proof to optimize correctness. Finally, there is a rich literature on the varied styles of (human) formal proofs (e.g., (Autexier & Dietrich, 2010; Wiedijk, 2004)). Our model, ImProver, builds on neural theorem proving techniques

---

[1]Code is available at https://github.com/riyazahuja/ImProver.

including full proof generation (Jiang et al., 2023; First et al., 2023), conditioning on example proofs (Jiang et al., 2023), retrieval (Yang et al., 2023; Thakur et al., 2024), and preceding file context (First et al., 2023; Hu et al., 2024), as well as error correction (Madaan et al., 2023; Chen et al., 2023) and documentation retrieval (Zhou et al., 2023) from code generation. ImProver brings these code generation techniques, along with new Chain-of-States prompting and meta-programmed contextual information, into a unified proof optimization agent.

# 3 AUTOMATED PROOF OPTIMIZATION WITH ImProver

Given a theorem statement $x$, additional context $c$, and an initial proof $y_0$, proof optimization consists of generating a new proof $y$ that is correct and minimizes (or maximizes) a metric $\mu(x, c, y_0, y) \to \mathbb{R}$.

## 3.1 METRICS

By varying the metric, we can perform tasks such as shortening proofs, making them more declarative in structure, or even automated proving. We consider four metrics:

*Length Metric:* The length metric measures the number of tactic invocations in the tactic proof, aiming to reduce the proof's length while ensuring its correctness. Note that shorter proofs often represent more efficient proofs.

*Declarative Metric:* We aim to rewrite proofs to be written in a declarative style (Autexier & Dietrich, 2010; Wiedijk, 2004), which is related to the number of independent subproofs in a proof. Intuitively, this corresponds with a sense of structure for the proof, and can be interpreted as being more readable, explicit, or modular in style. Concretely, we evaluate declarativity using the ratio of number of explicitly typed `have` tactics to total number of tactic invocations.

*Mixed Metric:* We aim to combine the length and declarative metrics as described above to generate proofs that are both concise and declarative. This is done by penalizing all tactics, and rewarding declarative tactics (e.g. `have`) in order to amortize the net total. We then aim to maximize the net score. In practice, we assign a value of $-1$ to each tactic in the proof, and a value of $+5$ for declarative tactics, meaning that each declarative tactic "pays" for $4$ additional tactics.

*Completion Metric:* The completion of a proof simply describes its correctness. This is a trivial metric which measures the number of errors present. The completion metric is used for concretely viewing proof optimization as a generalization of neural theorem proving.

**Degenerate Solutions.** Our goal here has been to provide a flexible means to optimize proofs with respect to any metric that might prove useful. The particular metrics we use here are intentionally simplistic, in that they are used only to test and evaluate the method. The task of designing metrics that correspond more accurately to human criteria or are optimal for various training tasks is left to later work as what defines a good metric are dependent on the use case.

Additionally, we note the possibility of degenerate solutions, as in, generations of proofs that score highly on a certain metric, while not corresponding to the intuitive sense of that metric. For example, overuse of `have` statements can greatly increase the declarativity of the proof, despite not being used in the proof's deductive process whatsoever. It is undesirable for a model to generate such degenerate solutions, and to account for this, we guide the model with many human-written examples of each metric in question, rather than requiring it to solely maximize a reward function. For more complex, user-defined metrics, the possibilities for degenerate solutions only increases, and as such, guiding models using concrete examples as well as using reward models rather than reward functions to score metrics may mitigate the risks of such degenerate solutions.

## 3.2 IMPROVER

We develop several improvements that can be applied to a black-box LLM generator $y_{out} \sim G(\cdot|x_{in})$, such as GPT-4 (OpenAI et al., 2024), and specify ImProver with respect to these parameters. The explicit prompts and templates that are sent to the LLM can be found in (§A).

**Without Chain-of-States**

```
example : s ∩ t ∪ s ∩ u ⊆ s ∩ (t ∪ u)  := by
  rintro x (⟨xs, xt⟩ | ⟨xs, xu⟩)
  · use xs; left; exact xt
  . use xs; right; exact xu
```

**With Chain-of-States**

```
example : s ∩ t ∪ s ∩ u ⊆ s ∩ (t ∪ u)  := by
  rintro x (⟨xs, xt⟩ | ⟨xs, xu⟩)
  /-
  case inl.intro
  α : Type u_1
  s t u : Set α
  x : α
  xs : x ∈ s
  xt : x ∈ t
  ⊢ x ∈ s ∩ (t ∪ u)
  case inr.intro
  α : Type u_1
  s t u : Set α
  x : α
  xs : x ∈ s
  xu : x ∈ u
  ⊢ x ∈ s ∩ (t ∪ u)
  -/
  · use xs; left; exact xt
  /-
  Goals Solved!
  -/
  . use xs; right; exact xu
  /-
  Goals Solved!
  -/
```

Figure 2: A Lean proof (left) with Chain-of-States prompting annotations (right).

### 3.2.1 CHAIN-OF-STATES PROMPTING

Typical formal proofs are a sequence of tactics (akin to steps) and *states* that show the hypotheses and goals at each step. The intermediate states often contain valuable information (e.g., an expression after it has been simplified) that is not present in the tactics. To allow the model to reason about these intermediate goals and hypotheses, we use tools from Lean metaprogramming to automatically annotate each proof state as a comment prior to each tactic. We refer to this method as *Chain-of-States* (CoS) prompting since it makes intermediate states explicit, akin to how chain-of-thought prompting (Wei et al., 2022) makes intermediate steps of a solution explicit.

These states are extracted directly and symbolically from the underlying Lean compilation steps using Lean's rich metaprogramming suite. The implementation of this extraction system is modeled from the work (Kim Morrison, 2024). Specifically, in the compiler's elaboration and evaluation stages – where the parsed theorem code is first converted into concrete syntax trees (in practice, Syntax objects) and abstract syntax trees (Expr objects) – we convert the CST and AST output objects into the relevant proof data and proof states in the form of proof trees (Lean.Elab.InfoTree). These proof trees contain detailed context and information on a tactic-by-tactic level relating to the modification of the proof state, metavariable context, and proof correctness.

After state extraction is completed and cached for efficient future access, we annotate the proof text itself to contain the intermediate states in the form as comments. Figure 2 shows an example.

This explicit reasoning aims to help the generator model construct more optimized proofs via additional symbolic data.

### 3.2.2 OUTPUT FORMATTING.

LLM outputs often contain ancillary and syntactically invalid content, especially before and after the actual proof. Additionally, by applying additional structure to the LLM outputs, we may hope to generate more structured proofs. To analyze this hypothesis, we introduce two additional output formats in addition to the standard string output: string list and string tree. The former enforces the model output of a proof to be a tactic sequence represented as a list of strings, and the latter enforces proofs to be written as proof trees, represented as a tree of strings.

### 3.2.3 SAMPLING METHOD

We also introduce different methods of sampling between many (sequential or parallel) LLM inference calls, involving best-of-n and iterative refinement implementations, as well as combinations thereof.

**Best-of-n.** The best-of-n technique generates multiple ($n$) calls to the language model and selects the "best" via a simple selection policy that first prioritizes output correctness, and secondly prioritizes the evaluated metric delta score.

Using a temperature value of $1$, we ensure that our $n$ calls to the model are diverse, as the temperature hyperparameter (ranging between $0$ and $2$) controls the randomness of the outputs. The default value of $1$ ensures that outputs are sufficiently random without sacrificing accuracy and generating unpredictable behavior. Moreover, this allows for sufficient variance that the best-of-n scoring function has many distinct inputs to choose from.

More specifically, our scoring function is given by the 2-ary comparison function $S$, whose arguments are output objects $y, y'$.

$$S(y, y') = \begin{cases} \max(y, y', \text{key: } x \mapsto \mu(x)), & E(y) = E(y') = 0 \\ y, & E(y) = 0, E(y') > 0 \\ y', & E(y) > 0, E(y') = 0 \\ \min(y, y', \text{key: } x \mapsto E(x)), & E(y) = E(y') > 0 \end{cases}$$

Where $\mu(x)$ is the metric score of $x$, and $E(x)$ is the number of errors in $x$. This comparison function can be extended to evaluate the best output of any finite $n$ via induction.

**Error correction and Refinement.** Inspired by self-debugging techniques in code generation (Madaan et al., 2023; Chen et al., 2023), ImProver identifies and corrects errors in the generated proofs by iteratively refining its outputs. The refinement process relies on user-defined parameters `n` and `prev_num` to specify the number of iterations and the number of previous iterations' data to forward, respectively. Each iteration carries information on the last `prev_num` iterations, including input, output, metric score, correctness, and error messages.

**Combination Sampling and Compound Prompt Functions.** Compound prompt functions utilize the curried nature of the back-end implementations of best-of-n and refinement to nest these techniques within one another. For example:

`best_of_n((refinement,m),n)` is a compound sampling method that run a best-of-$n$, where each call is a $m$-step refinement.

`refinement((best_of_n,m),n)` is a compound sampling method that runs a $n$-step refinement, where each call is a best-of-$m$ call to the LLM.

Note that with each of these compound prompt functions, there are always a total of $mn$ iterations.

### 3.2.4 RETRIEVAL

ImProver uses MMR (Maximum Marginal Relevance)-based (Carbonell & Goldstein, 1998) retrieval-augmented generation to select relevant examples and documents. More specifically, for a user-specified $k$, example retrieval selects the $k$ most relevant examples of proof optimization on a specific metric. additionally, document retrieval extracts information using MMR from a pair of fixed (vector) databases for the specified metric. The databases store syntactically chunked data from the Theorem Proving in Lean (TPiL) handbook – containing syntax guides and tactic explanations – and the Mathlib mathematics libary – containing thousands of theorems and lemmas.

The Mathlib retriever finds the top $k$ documents that score the highest MMR score against the current theorem, the TPiL retriever finds the top $k$ documents that score the highest MMR score against the current theorem in context and all current error messages. This retrieval process helps in generating more contextually accurate prompts that allow the language model to better correct its own errors as well as find useful lemmas to reference.

## 4 EXPERIMENTS

We test ImProver on rewriting real-world undergraduate theorems, competition problems, and research-level mathematics and compare its results to those of the base GPT-4o and GPT-4o-mini models. We examine the optimization capabilities of ImProver for the length and declarative metrics,

studying the effectiveness in maintaining the correctness of the tactic proof while making it more concise as well as making it more declarative in style and structure.

## 4.1 SETUP

Our experimentation is split into three distinct stages. We first perform ablation testing on the ImProver model parameters (§3.2) to ensure that ImProver's parameter specification is the optimal one with respect to correctness and metric optimization score. We then evaluate this optimal parameter combination on datasets of varying complexity and analyze the performance and results thereof. Lastly, we note the performance of ImProver in NTP applications in comparison to the base GPT-4o and GPT-4o-mini models.

**Datasets.** We evaluate ImProver on subsets of the Mathematics in Lean (MIL) (leanprover-community, 2024), Compfiles (David Renshaw, 2024), and Mathlib (mathlib Community, 2020) datasets. Additionally, ablations are performed on a subset of MIL, and theorem proving is benchmarked on various subsets of MIL as well as the MiniF2F (Zheng et al., 2022) dataset. Details of the datasets used in each experiment is included in appendix B.1.

**Models.** Our base generator uses GPT-4o (OpenAI et al., 2024) (`gpt-4o-2024-08-06`). Since no prior methods currently exist for automated proof optimization, we consider a prompted GPT-4o without the improvements described in (§3.2) as our baseline. Additionally, the baseline and ImProver both receive a prompt containing instructions to optimize for the given metric, with the theorem statement, context, and initial proof. ImProver augments this prompt with the data from the improvements described in §3.2. Additional input information is detailed in appendix A.

**Performance metrics.** Since proof optimization is a new task, we define four performance metrics for measuring aspects of correctness and improvement.

First, we define *improvement* for length as percentage change in length, $\frac{\mu_{\text{len}}(y_0) - \mu_{\text{len}}(y)}{\mu_{\text{len}}(y_0)} \times 100$. For readability, we use the difference, $\mu_{\text{read}}(y) - \mu_{\text{read}}(y_o)$. If no correct output is generated by the model for a specific theorem, improvement is defined to be zero. We define *nonempty improvement* as the improvement restricted to theorems for which some output has nonzero improvement. Intuitively, improvement is the expected improvement in metric score from the input to output, accounting for errors in the generation. The nonempty improvement score is the expected improvement in metric score, given that there are no errors in the generation.

Additionally, the *accuracy* is the percentage of theorems in the dataset which the model was able to generate a correct output for. The *improved accuracy* is the percentage of theorems in the dataset which the model was able to generate a correct output for, as well as improve the metric to be nonzero.

### 4.1.1 ABLATION SETUP

When performing our ablation studies, we used a fixed dataset (MIL; see appendix B.1) and metric (length) and varied the parameters of all the features to find the optimal combination. However, as there are over 8640 possible combinations, rather than test all combinations, we evaluate using a factorial testing method.

**Testing Groups.** We define the following testing groups with the specified parameter combinations:

*GPT-4o-mini/GPT-4o:* This varies the GPT-4o model, outputting a `string` with no other features.

*Output and CoS:* We evaluate the effects of different output formatting styles (`string`, `string list`, `string tree`) and CoS (True, False), with the model fixed as GPT-4o, with no other features enabled.

*Example Retrieval:* We evaluate the effects of increasing the number of examples provided (multi-shot prompting) in the range of $0, 3, 5, 7$, and $10$, with the model fixed as GPT-4o, CoS and output formatting fixed as the best combination from the previous test, and no other features enabled.

*Sampling Method:* Here, we evaluate the effects of best-of-n and refinement for a fixed $n = 5$. Additionally we test on the refinement cases if forwarding the most recent iteration result, or all previous iteration results is the best, and if we should keep the best out of the iterations, or the most

Table 1: Average Proof optimization results.

| Metric | Model | Improvement | Nonempty Improvement | Accuracy | Improved Acc. |
|--------|-------|-------------|----------------------|----------|---------------|
| **Length** | GPT-4o | 3.7 | 15.15 | 26.36% | 8.31% |
| | ImProver | **20.96** | **55.29** | **100.0%** | **35.44%** |
| **Declarativity** | GPT-4o | 2.21 | 8.02 | 18.75% | 6.13 % |
| | ImProver | **9.34** | **30.53** | **100.0%** | **24.56%** |
| **Mixed** | GPT-4o | 3.51 | 23.90 | 14.70% | 5.11% |
| | ImProver | **27.31** | **39.24** | **100.0%** | **30.55** |

Table 2: MIL Proof optimization results.

| Metric | Model | Improvement | Nonempty Improvement | Accuracy | Improved Acc. |
|--------|-------|-------------|----------------------|----------|---------------|
| **Length** | GPT-4o | 6.25 | 18.58 | 37.5% | 14.42% |
| | ImProver | **30.54** | **56.56** | **100.0%** | **50.0%** |
| **Declarativity** | GPT-4o | 4.18 | 14.48 | 28.85% | 11.54% |
| | ImProver | **13.45** | **30.97** | **100.0%** | **34.21%** |
| **Mixed** | GPT-4o | 3.70 | 27.38 | 13.51% | 0.0% |
| | ImProver | **43.55** | **44.76** | **100.0%** | **45.94%** |

recent. The model is fixed as GPT-4o, CoS, output formatting, and examples are fixed as the best combination from the previous test, and no other features enabled.

*$n$ and Model:* Here, we evaluate the effects of larger $n$ values and different models. We test $n = 3, 5, 7, 10, 15$ on GPT-4o and GPT-4o-mini, as well as $n = 20$ for GPT-4o-mini (as it is of a far lower token cost). CoS, output formatting, examples, and sampling method are fixed as the best combination from the previous test, and no other features enabled.

*Combos and RAG:* We evaluate combination methods `refinement(best_of_m',m)` and `best_of_m'(refinement(m))`, for $m \neq m'$ with $mm'$ equal to the optimal value $m$ from the previous test. We also test the effect of enabling document retrieval. Model, CoS, output formatting, examples, $n$, and sampling method are fixed as the best combination from the previous test.

**Selection.** For each testing group, we select the best parameter combination — which is then held as constant for the testing of all future testing groups — based on the combination that has the maximal improvement score. This improvement score represents the expected improvement in metric score, accounting for possible errors in the generation; selecting the parameter combination with the highest such score allows for rewarding both generation accuracy and large improvements in the metric score.

Comparing this with the other three performance metrics, accuracy is not prefered as a selection heuristic, as by simply returning the initial input, we can get $100\%$ accuracy. Improved accuracy accounts for this by only counting theorems that has some positive improvement in metric score in the calculation, but this does not reward larger improvements to metric score any differently than smaller ones. Conversely, nonempty improvement ignores incorrect generations, so it is also not preferable for selection. The improvement score accounts for all this, rewarding correct generations and discouraging incorrect ones, and placing a higher weight to larger improvements in metric score.

**Ablation datasets.** We evaluate our ablations on a subset of MIL as detailed in appendix B.1.

## 4.2 RESULTS

ImProver **is capable of optimizing proofs in all settings.** From Table 2, Table 3, and Table 4, we can see that ImProver is capable of optimizing proofs on all datasets for both the length and declarative metrics, as well as on the mixed metric. Furthermore, Table 1 shows that across all metrics, ImProver significantly outperforms GPT-4o on proof optimization tasks on every experimental measure – aggregated from all datasets. Additionally, from Table 2, Table 3, and Table 4, we can see that ImProver outperforms GPT-4o on each dataset as well. We proceed to analyze this data and its implications.

Table 3: Compfiles Proof optimization results.

| Metric | Model | Improvement | Nonempty Improvement | Accuracy | Improved Acc. |
|---|---|---|---|---|---|
| **Length** | GPT-4o | 2.75 | 30.7 | 11.54% | 5.13% |
| | ImProver | **18.86** | **54.48** | **100.0%** | **34.62%** |
| **Declarativity** | GPT-4o | 0.39 | 3.38 | 14.1% | 1.28% |
| | ImProver | **5.74** | **24.89** | **100.0%** | **19.23%** |
| **Mixed** | GPT-4o | 3.96 | 3.96 | 26.9% | 20.0% |
| | ImProver | **20.60** | **76.53** | **100.0%** | **23.07%** |

Table 4: Mathlib Proof optimization results.

| Metric | Model | Improvement | Nonempty Improvement | Accuracy | Improved Acc. |
|---|---|---|---|---|---|
| **Length** | GPT-4o | 0.0 | 0.0 | 16.67% | 0.0% |
| | ImProver | **6.19** | **53.65** | **100.0%** | **11.54%** |
| **Declarativity** | GPT-4o | 0.0 | 0.0 | 4.65% | 0.0% |
| | ImProver | **4.63** | **33.19** | **100.0%** | **11.63%** |
| **Mixed** | GPT-4o | 2.92 | **30.14** | 9.30% | 4.65% |
| | ImProver | **4.16** | 7.45 | **100.0%** | **9.30%** |

**Length optimization.** First focusing on the length metric, we see that ImProver outperforms GPT-4o with respect to the improvement score by $566\%$ (aggregated over all datasets). Additionally, we are guaranteed that ImProver produces a correct output, although that output may just be the same as the input. However, $35.44\%$ of the time, it generates a correct output that is not the same length as the input, and in that case, we expect an average of a $55.29\%$ reduction in length. Comparing this with GPT-4o, we conclude that not only can ImProver optimize at a higher level on arbitrary theorems, but its ability to generate nontrivial correct outputs is far greater in comparison to GPT-4o.

**Declarativity optimization.** Declarativity optimization is similar, with ImProver outperforming GPT-4o by $423\%$. Moreover, the accuracy, improved accuracy, and nonempty improvement disparities for declarativity parallel those of the length tests. However, it should be noted that for both GPT-4o and ImProver, the accuracy and improved accuracy scores were markedly smaller for declarativity than length optimization. This suggests that for both models, it was generally more "difficult" to generate a correct output, and moreover, generate a correct output with a better metric score than the input, for declarativity optimization than length optimization. In other words, optimizing for declarativity is more difficult for the underlying generator than optimizing for length. However, we speculate with higher-quality prompts and metrics, this disparity can be minimized. Regardless, we note that different metrics can be less likely to be correctly optimized, and that model performance is correlated with the metric it seeks to optimize, both for GPT-4o and ImProver.

**Mixed optimization.** For the mixed optimization of both length and declarativity, we observe a $778\%$ outperformance by ImProver in the aggregate, with similar increases across all experimental measures. It is notable that the combined improvements and accuracies mirror the trends from the individual length and declarativity metrics, suggesting that ImProver is able to scale its optimizations to align with the increased complexity of the metric. Moreover, this empirically shows that for more complex and arbitrarily-defined metrics, ImProver maintains its ability to generate nontrivial optimizations, given sufficiently high-quality examples, prompts, and scoring functions.

**Optimization varies based on dataset difficulty.** Additionally noting Table 2, Table 3, and Table 4, we observe that the improvement score for all metrics for both GPT-4o and ImProver is highest for the MIL dataset, lower for Compfiles, and the lowest on the Mathlib theorems. This suggests that the expected improvement in metric score decreases with higher difficulty, with undergraduate-level theorems having a significantly higher expected improvement than research-level theorems. However, it should be noted that for all metrics, the nonempty improvement of ImProver stayed somewhat consistent, whereas for GPT-4o, it followed the aforementioned trend of decreasing with difficulty. Similarly, the accuracy and improved accuracy scores for both metrics and models decreased with higher difficulty datasets (disregarding ImProver's accuracy scores, as they are ensured to be 100%). This suggests that although the base GPT-4o generator is less likely to generate a correct output for

higher difficulty datasets, the improvements that ImProver makes to the base generator allows it to maintain its improvement in the metric score whenever a correct output is generated. As such, we can speculate that the bottleneck in the improvement score is not the model's ability to optimize the proof for a metric, but rather its ability to generate a new correct proof at all. As such, we conjecture that with more capable generator models, the accuracy — and thus, the improvement score — in optimization tasks will continue to increase, until the improvement scores match the nonempty improvement.

Overall, we conclude that although the performance of both ImProver and GPT-4o decreases on length and declarativity optimization on more difficult datasets, ImProver significantly outperforms GPT-4o on all datasets for length, declarativity, and mixed optimization.

### 4.2.1 Ablation Testing

Table 5: Ablation results. Each cell in the ablation tests shows `best` / `worst`, which are the `best` and `worst` parameter combinations in the test group.

|                      | Improvement    | Nonempty Improve. | Accuracy          | Improved Acc.       |
| -------------------- | -------------- | ----------------- | ----------------- | ------------------- |
| GPT-4o-mini          | 0              | 0                 | 3.62%             | 0%                  |
| GPT-4o               | 7.03           | 19.67             | 35.77%            | 15.33%              |
| + Output and CoS     | 8.04 / 6.31    | 12.38 / 14.17     | 64.96% / 44.53%   | 21.17% / 16.06%     |
| + Example Retrieval  | 9.34 / 5.67    | 14.7 / 8.44       | 63.5% / 67.15%    | 21.9% / 16.79%      |
| + Sampling Method    | 15.35 / 9.34   | 18.44 / 14.7      | 83.21% / 63.5%    | 36.5% / 21.9%       |
| + $n$ and Model      | 23.51 / 3.65   | 26.28 / 4.63      | 89.47% / 78.95%   | 45.61% / 8.77%      |
| + Combos and RAG     | 34.88 / 28.25  | 57.56 / 33.48     | 60.61% / 84.38%   | 54.55% / 53.12%     |
| ImProver             | **34.88**      | **57.56**         | **100%**          | **54.55%**          |

We perform ablation studies using a subset of the MIL dataset as discussed in §4.1.1. The results of this factorial study are aggregated in Table 5. We measure the baseline results from the GPT-4o and GPT-4o-mini models, noting that GPT-4o is the better-scoring model (with respect to the improvement score). Thus, fixing this model, we vary the output formatting type and if CoS is enabled, and determine that outputting `string list` with CoS enabled maximizes the improvement score. Fixing these parameters, we now vary the number of examples retrieved, noting that prompting with 10 examples maximizes the improvement score. Fixing this parameter, we vary the sampling methods (excluding compound methods and fixing $n = 5$) and observe that best-of-n is the best parameter combination. Now, as GPT-4o-mini is significantly less computationally expensive than its GPT-4o counterpart, we test both models with the sample method fixed to best-of-n, and vary $n = 1, 3, 5, 7, 10, 15$, and for GPT-4o-mini, also $n = 20$. We conclude that GPT-4o with $n = 15$ is the most effective. Fixing these parameters, we consider all mixed compound sampling methods with and without document retrieval enabled, concluding that a 5-step refinement with best-of-3 on each iteration, with RAG enabled, is the optimal combination.

Thus, as we can see from Table 5, the optimal parameter combination comes from gpt-4o outputting as a `string list` with CoS, RAG, 10 examples, 5-step refinement with each iteration being a best-of-3 evaluation. Changing any one of these parameters them leads to a reduction in performance. Additional ablation data can be found at (§B.2).

**Declarativity and Chain-of-States (CoS) Ablation.** We additionally examine the effects of disabling CoS on declarativity optimization tasks, as we speculate that CoS has a high impact on the performance of declarativity optimization tasks, as the proof states that are embedded due to CoS seem to be a critical aspect to generating the explicit declarations that the declarative metric measures.

We confirm this result by considering Table 6 and observe that enabling CoS nearly doubles the improvement score, and significantly improves the nonempty improvement score, suggesting that CoS has a large impact on optimizing for the declarative metric, as conjectured. However, we also note a significant increase in improved accuracy, which suggests that embedding the chain of states also improves the ability of the model to generate nontrivial correct outputs, implying that the symbolic information contained in the states are critical to effectively making a proof more declarative.

**Syntax Guidance Ablation.** We examine the effects of syntax guidance on ImProver's performance. To test this, we consider a subset of MIL (B.1), and optimize for length with and without error

Table 6: CoS Declarativity Ablation results.

|  | Improvement | Nonempty Improve. | Accuracy | Improved Acc. |
|---|---|---|---|---|
| GPT-4o | 4.97 | 15.89 | 37.5% | 12.5% |
| ImProver, CoS Disabled | 9.23 | 24.61 | 100.0% | 28.12% |
| ImProver | **16.69** | **31.42** | **100.0%** | **46.88%** |

Table 7: Syntax Guidance Ablation results.

|  | Improvement | Nonempty Improve. | Accuracy | Improved Acc. |
|---|---|---|---|---|
| GPT-4o | 11.00 | 25.94 | 42.42% | 21.21% |
| ImProver, No Syntax Guidance | 23.42 | 49.97 | 100.0% | 46.88% |
| ImProver | **28.94** | **48.74** | **100.0%** | **59.38%** |

message forwarding. Considering the results of this ablation in Table 7, we observe that without syntax guidance and error forwarding, the ability of the model to improve the metric score is approximately unchanged, but there is a significant 13% spike in improved accuracy. This signifies that the syntax guidance improves the model's ability to generate correct results – as is expected – but does not improve the model's ability to optimize proofs assuming correct generations. This ensures that the large improvement in performance compared to GPT-4o is not solely due to simple syntax guidance, but moreso caused by improvements like CoS, example retrieval, retrieval, etc.

### 4.2.2 NEURAL THEOREM PROVING EVALUATION

Table 8: Proof generation results. Each cell shows percent accuracy.

|  | MIL-C04 Pass@15 | MIL-C08 Pass@15 | MIL Pass@15 | MiniF2F-test Pass@8 |
|---|---|---|---|---|
| GPT-4o | 18.18% | 25% | 21.73% | 9.02% |
| ImProver | **45.45%** | **33.33%** | **39.13%** | 16.39% |
| Lean Expert Iteration | - | - | - | **34.5%** |

We evaluate ImProver's neural theorem proving (NTP) performance using the completion metric on a subset from MIL with empty input proofs (B.1).Table 8 shows the accuracy on the dataset split by topic for both ImProver and GPT-4o. ImProver substantially outperforms GPT-4o across all datasets. Thus, proof optimization systems do indeed generalize NTP systems.

Additionally, we note that specialized neural theorem proving models such as GPT-f (Zheng et al., 2022) and Lean Expert Iteration (Polu et al., 2022a) outperform ImProver, but as they are specially trained on a large corpus of Lean code (whereas ImProver is an extension of GPT-4o), as well as designed for theorem proving rather than proof optimization, it is worthwhile for future works to consider fine-tuning or specializing ImProver's methodology specifically for theorem proving. We reiterate that the purpose of this experiment is to empirically prove that proof optimization systems like ImProver do indeed generalize the problem of theorem proving as claimed in §3.1.

## 5 CONCLUSION

In this paper, we introduced ImProver, a novel agent-based tool for automated proof optimization in Lean. By incorporating CoS, RAG, and other features, ImProver significantly outperforms base language models in proof optimization over undergraduate, competition, and research-level problems.

However, ImProver is limited by its high cost, which is exacerbated by its reliance on proprietary LLM's. In future work, we will apply SFT and RL on a smaller model to match performance locally.

ImProver demonstrates its ability to generate substantially shorter and more declarative proofs while maintaining correctness. As such, we believe that ImProver sets the stage for further work on proof optimization to advance the study and use of AI in mathematics.

### ACKNOWLEDGEMENTS

We thank the L3 Lab, Hoskinson Center for Formal Mathematics, Convergent Research, Lean FRO, and the OpenAI Researcher Access Program for their support.

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

# A    PROMPTS

In this appendix, we note the prompts used by ImProver both for general LLM prompting, as well as the metric-specific prompts.

## A.1    TEMPLATE

For the main prompt sent to the LLM on each sample, we build a prompt string using a chat prompt template that is then invoked at runtime to fill in the variables.

Namely, these variables include the set of metric prompts, previous results, input theorem, context, a syntax documents, Mathlib documents, and examples.

The prompt template is a conversation of the format:

> **Placeholder:** *All metric prompts with a 'System' role*
>
> **System:** You will be given the proof context (i.e. the lean file contents/imports leading up to the theorem declaration) wrapped by <CONTEXT>...</CONTEXT>.
> You will be given the previous *num_prev* input/output pairs as well as their metric (metric.name) score and correctness score, as well as any error messages, for your reference to improve upon. Each of these previous results will be wrapped with <PREV I=0></PREV I=0>,...,<PREV I=*num_prev-1*></PREV I=*num_prev-1*>, with I=*num_prev-1* being the most recent result.
> Remember to use lean 4 syntax, which has significant changes from the lean 3 syntax. To assist with the syntax relating to the current theorem and current error messages, you will be given *num_syntax_docs* documents to refer to for fixing these syntax issues. Each of these documents will be wrapped with <SYNTAX_DOC>...</SYNTAX_DOC>.
> You will also receive *num_mathlib_docs* documents relevant to the current theorem to help with formulating your modified proof. Each of these will be wrapped with <CONTENT_DOC>...</CONTENT_DOC>
> You will also receive *num_examples* examples of input-output pairs of proofs that were optimized for the *metric* metric. Each of these will be wrapped with <EXAMPLE>...</EXAMPLE>
> You will be given the tactic states as comments for reference. The current theorem will be wrapped in <CURRENT>...</CURRENT>
>
> **System:** *Output format instructions*
>
> **Placeholder:** *All retrieved syntax documentation*
>
> **Placeholder:** *All retrieved mathlib documentation*
>
> **Placeholder:** *All retrieved examples*
>
> **User:** <CONTEXT> *context* </CONTEXT>
>
> **Placeholder:** *Previous results and inputs/outputs*
>
> **Placeholder:** *All metric prompts with a 'User' role*
>
> **User:** <CURRENT> *theorem* </CURRENT>

This prompt is then invoked and sent to the language model by filling in all the variables and placeholders. Notably, when we invoke the chain given by `chain|llm|parser`, we throttle the invocation with a randomized exponential rate limit throttling to account for API rate limits, especially in highly-parallelized requests like when benchmarking over a large number of theorems.

## A.2    METRIC PROMPTS

**Length Metric**

> **System:** You are an AI assistant who shortens Lean 4 proofs while ensuring their correctness. You will aim to reduce the number of lines of the tactic proof while ensuring that it properly compiles in Lean 4.

**User:** Shorten the current theorem (wrapped in <CURRENT>...</CURRENT>) to be as short in length—measured in the number of lines of the proof—as possible, while also ensuring that the output is still syntactically correct."

**Declarativity Metric**

**System:** You are an AI assistant who rewrites Lean 4 proofs to be more readable while ensuring their correctness. We measure readablity by considering the ratio of the number of explicitly typed `have` tactics against the total number of tactics in the proof, as this is proportional to whether a proof is declarative in style, and thus, readable.

**User:** Rewrite the current theorem (wrapped in <CURRENT>...</CURRENT>) so it is more readable and declarative and modular.

**Mixed Metric**

**System:** You are an AI assistant who rewrites Lean 4 proofs to be higher quality, namely, more concise and more readable/declarative in style and structure. We measure the length of a proof by the number of tactics, and readability/declarativity by the number of explicitly typed `have` tactics.

**User:** Rewrite the current theorem (wrapped in <CURRENT>...</CURRENT>) so it is more readable and declarative and concise, while also being correct. Namely, we penalize 1 point for every tactic, and reward 5 points for every declarative tactic (namely, `have` statements). Your goal is to maximize that reward function as much as possible while generating a correct proof using the provided template as a starting point.

**Completion Metric**

**System:** You are an AI assistant who automatically solves Lean 4 proofs (as in, generates the tactic proof) and ensures its correctness. You will receive a Lean 4 proof you must modify to eliminate any errors so that it compiles correctly and eliminate any "sorry"s with full proofs.

**User:** Rewrite the current theorem (wrapped in <CURRENT>...</CURRENT>) so it is a formal, complete, and correct Lean 4 proof by filling in its tactic proof.

## A.3 METRIC EXAMPLES

In this section, we illustrate side-by-side examples of metric optimization. These examples are part of a larger set of examples provided to the model as described in §*A.1*.

**Length Metric** As shown in Figure 3, we provide the model an example of using more advanced tactics like `rintro` and inlining `apply` statements to shorten the proof from 5 tactics to 2.

**Suboptimal**

```
example : (P → Q) ∧ (Q → R) → P → R := by
  intro h p
  rcases h with ⟨a,b⟩
  apply b
  apply a
  exact p
```

**Length Optimized**

```
example : (P → Q) ∧ (Q → R) → P → R := by
  rintro (⟨hpq,hqr⟩) hp
  exact hqr (hpq hp)
```

Figure 3: A human-written example of length optimization.

**Declarative Metric**

As shown in Figure 4, we provide the model an example of adding an intermediate result `hp_nq` with an explicitly written type of P → ¬Q. Additionally, we show the model an example of simplifying tactics and external lemmas and dependencies to solve the problem in a more direct, declarative, and readable manner.

**Mixed Metric**

**Suboptimal**

```
example (h : ¬ (P ∧ Q)) : ¬ P ∨ ¬ Q := by
  push_neg at h
  exact not_or_of_imp h
```

**Declarativity Optimized**

```
example (h : ¬ (P ∧ Q)) : ¬ P ∨ ¬ Q := by
  have hp_nq : P → ¬ Q := by
    intro p q
    exact h ⟨p,q⟩
  by_cases hp:P
  . right
    exact hp_nq hp
  . left
    exact hp
```

Figure 4: A human-written example of declarativity optimization.

As shown in Figure 5, we provide the model an example of adding multiple intermediate results as well as using proof terms and structural modifications to shorten the number of tactics, resulting in a more optimized proof.

We note that although the "Suboptimal" example is taken from MIL C04, it is ensured to not be in our testing split to avoid data contamination.

**Suboptimal**

```
example : s \ t ∪ t \ s = (s ∪ t) \ (s ∩ t)
      := by
  ext x; constructor
  · rintro (⟨xs, xnt⟩ | ⟨xt, xns⟩)
    · constructor
      left
      exact xs
      rintro ⟨_, xt⟩
      contradiction
    . constructor
      right
      exact xt
      rintro ⟨xs, _⟩
      contradiction
  rintro ⟨xs | xt, nxst⟩
  · left
    use xs
    intro xt
    apply nxst
    constructor <;> assumption
  . right; use xt; intro xs
    apply nxst
    constructor <;> assumption
```

**Joint Length/Readability Optimized**

```
example : s \ t ∪ t \ s = (s ∪ t) \ (s ∩ t)
          := by
  ext x
  constructor
  have in_union_not_inter : ∀ x, x ∈ s ∪ t →
    x ∉ s ∩ t → x ∈ (s ∪ t) \ (s ∩ t) := by
    intro x hx hnx
    exact ⟨hx, hnx⟩
  have in_diff_union : ∀ x, (x ∈ s \ t ∨ x ∈
    t \ s) → x ∈ (s ∪ t) \ (s ∩ t) := by
    rintro x (⟨xs, xnt⟩ | ⟨xt, xns⟩)
    . exact in_union_not_inter x (Or.inl xs)
      (fun ⟨_, ht⟩ => xnt ht)
    . exact in_union_not_inter x (Or.inr xt)
      (fun ⟨hs, _⟩ => xns hs)
  exact in_diff_union x
  have in_diff_or : ∀ x, x ∈ (s ∪ t) \ (s ∩
    t) → (x ∈ s \ t ∨ x ∈ t \ s) := by
    rintro x ⟨hx, hnx⟩
    by_cases xs : x ∈ s
    . left; exact ⟨xs, fun ht => hnx ⟨xs, ht⟩⟩
    . right; exact ⟨hx.resolve_left xs, fun
      hs => hnx ⟨hs, hx.resolve_left xs⟩⟩
  exact in_diff_or x
```

Figure 5: A human-written example of mixed length/declarativity optimization optimization.

**Completion Metric**

As shown in Figure 6, we provide the model an example of showing a property about Set's, an externally defined datastructure, using simple tactics and forward reasoning, without external lemmas.

**Suboptimal**

```
example {α : Type*} (s : Set α) : s ∩ s = s
      := by
  sorry
```

**Completion Optimized**

```
example {α : Type*} (s : Set α) : s ∩ s = s
      := by
  ext x
  constructor
  . intro h
    rcases h with ⟨hs,_⟩
    exact hs
  . intro h
    constructor
    . exact h
    . exact h
```

Figure 6: A human-written example of proof completion.

## B    ADDITIONAL EXPERIMENTAL RESULTS

In this section, we provide more detailed information on the experimental setup and results used to evaluate ImProver.

### B.1    DATASET DETAILS

**Main Datasets** We evaluate our experiments on subsets of the following datasets:

*Mathematics in Lean (MIL) (leanprover-community, 2024):*   this dataset contains pedagogical solutions of common undergraduate-level exercises, and as such contains many declarative, yet verbose and inefficient proofs. We use exercise solutions from set theory, elementary number theory, group theory, topology, differential calculus, and integration & measure theory. This dataset contains theorems at an undergraduate-level of complexity. For our main results, we evaluated on 72 theorems from exercise solutions from MIL chapters $4, 5, 8, 9$, and $10$.

*Compfiles (David Renshaw, 2024):*   Solutions of International Mathematics Olympiad (IMO) and American Mathematics Olympiad (USAMO) competition problems from 2016 to 2024. This is a dataset of internationally-renowned competitive math problems, many of which are readable and declarative, yet quite verbose. This dataset contains theorems of a competitive format, and although they contain concepts only at a high-school level, the logical complexity of internationally-renowned competition results is far above that. For our main results, we used all $26$ theorems and lemmas from the Compfiles database of complete solutions to the International Mathematics Olympiad (IMO) and the American Mathematics Olympiad (USAMO) from 2016-2024.

*Mathlib (mathlib Community, 2020):* Mathlib contains many advanced results at the forefront of mathematics, and has been at the center of research-level formalizations. These proofs are concise and generalized - which often comes at the cost of readability, declarativity, and understandability. These results and theorems often are at the cutting edge of research and a highest level of complexity compared the the other two datasets.

For our main results, we evaluated our methods on $43$ advanced research-level proofs from `Mathlib/AlgebraicTopology/FundamentalGroupoid`. This is the most difficult dataset.

**Ablation Datasets**

We evaluate our ablations on a subset of MIL. Additional details on this subset is included in appendix B.1.However, due to the increase in model calls for larger $n$ values, we switch a representative sample of this subset for some test groups. Namely,

GPT-4o-mini, GPT-4o, Output and Cos, Example Retrieval, and Sampling Method are tested on the $133$ theorems in the solutions of `C03_Logic`, `C04_Sets_and_Functions`, and `C05_Elementary_Number_Theory`.

$n$ and Model are tested on $55$ theorems from a representative sample of the aforementioned, and Combos and RAG are tested on a representative sample of $32$ theorems from the aforementioned.

Additionally, we note that both the Declarativity/CoS ablation and the Syntax Guidance ablation are performed on the same $32$ theorems sample as mentioned above.

**Completion Datasets**

We initially evaluate our completion/NTP dataset on 23 exercises from Mathematics in Lean. Namely, we consider a representative sample of 12 exercises in group theory (Chapter 8; denoted "MIL-C08", done at 15 samples), 11 exercises in set theory (Chapter 4; denoted "MIL-C04", done at 15 samples). Moreover, we ensure that all these theorems have an empty proof.

Additionally, we evaluate on the MiniF2F-test (Zheng et al., 2022) dataset with $8$ samples (where ImProver runs the samples as 2 refinement steps of a best-of-4 call each).

This experiment is intended to be an initial evaluation to show that automated proof optimization systems can generalize neural theorem proving, however, future work will explore the effects of policy optimization and fine-tuning of ImProver's methodology to perform neural theorem proving more competitively against specialized NTP models.

## B.2 ABLATION DETAILS

We now proceed to show detailed results from our ablation testing.

Table 9: Output and Chain-of-States Ablations

| Output Format | CoS | Improvement | Nonempty Improve. | Accuracy | Improved Acc. |
|---|---|---|---|---|---|
| `string` | True | 7.53 | 16.12 | 46.72% | 16.79% |
| `string` | False | 7.03 | 19.67 | 35.77% | 15.33% |
| `string list` | **True** | **8.04** | **12.38** | **64.96%** | **21.17%** |
| `string list` | False | 7.04 | 13.58 | 51.82% | 18.98% |
| `string tree` | True | 7.62 | 15.34 | 49.64% | 18.25% |
| `string tree` | False | 6.31 | 14.17 | 44.53% | 16.06% |

By Table 9, we see that the optimal combination in this testing group is a `string list` output format with CoS enabled. Fix these values for all future tests.

Table 10: Example Retrieval Ablations

| Examples | Improvement | Nonempty Improve. | Accuracy | Improved Acc. |
|---|---|---|---|---|
| 0 | 5.67 | 8.44 | 67.15% | 16.79% |
| 3 | 8.49 | 13.68 | 62.04% | 19.71% |
| 5 | 8.38 | 12.9 | 64.96% | 21.17% |
| 7 | 7.56 | 12.04 | 62.77% | 19.71% |
| **10** | **9.34** | **14.7** | **63.5%** | **21.9%** |

With the previous optimal parameters fixed, run the ablation on the number of examples. By Table 10, we see that the optimal combination in this testing group is 10 examples. Fix this value for all future tests.

Table 11: Sampling Method Ablations

| Method | Forward | Keep Best | Improvement | Nonempty Improve. | Accuracy | Improved Acc. |
|---|---|---|---|---|---|---|
| None | N/A | N/A | 9.34 | 14.7 | 63.5% | 21.9% |
| refinement | 1 | False | 14.76 | 30.63 | 48.18% | 30.66% |
| refinement | 5 | False | 12.5 | 20.88 | 59.85% | 30.66% |
| refinement | 1 | True | 14.95 | 14.95 | 100.0% | 30.66% |
| refinement | 5 | True | 13.15 | 13.15 | 100.0% | 29.93% |
| **best-of-n** | N/A | N/A | **15.35** | **18.44** | **83.21%** | **36.5%** |

Note that forward and keep-best values are parameters for refinement of how many previous iterations to forward, and whether to keep the most recent or the best iteration in subsequent refinement steps.

Now, with the previous optimal parameters fixed, run the ablation on the sample method. By Table 11, we see that the optimal combination in this testing group is best-of-n. Fix this value for all future tests.

With the previous optimal parameters fixed, run the ablation on the value of $n$ and model. By Table 12, we see that the optimal combination in this testing group is GPT-4o with $n = 15$. Fix this value for all future tests.

With the previous optimal parameters fixed, run the ablation on the combination methods and if RAG is enabled. By Table 13, we see that the optimal combination in this testing group is a 5-step refinement with each iteration being a best-of-3 call, with RAG enabled.

## B.3 ADDITIONAL QUALITATIVE EXAMPLES

In this section, we provide additional qualitative examples demonstrating the improvements ImProver achieves in proof optimization.

Table 12: Model and $n$ Ablations

| Model | $n$ | Improvement | Nonempty Improve. | Accuracy | Improved Acc. |
|---|---|---|---|---|---|
| gpt-4o | 3 | 19.66 | 24.36 | 80.7% | 38.6% |
| gpt-4o | 5 | 20.12 | 24.97 | 80.56% | 36.11% |
| gpt-4o | 7 | 22.44 | 27.21 | 82.46% | 42.11% |
| gpt-4o | 10 | 21.73 | 25.28 | 85.96% | 40.35% |
| **gpt-4o** | **15** | **23.51** | **26.28** | **89.47%** | **45.61%** |
| gpt-4o-mini | 3 | 3.65 | 4.63 | 78.95% | 8.77% |
| gpt-4o-mini | 5 | 5.12 | 6.21 | 82.46% | 10.53% |
| gpt-4o-mini | 7 | 3.65 | 4.34 | 84.21% | 8.77% |
| gpt-4o-mini | 10 | 4.99 | 5.69 | 87.72% | 12.28% |
| gpt-4o-mini | 15 | 4.35 | 5.06 | 85.96% | 12.28% |
| gpt-4o-mini | 20 | 4.87 | 5.56 | 87.72% | 14.04% |

Table 13: RAG and Combination Sampling Method Ablations

| Combination | $m$ | $m'$ | RAG | Improvement | Nonempty Improve. | Accuracy | Improved Acc. |
|---|---|---|---|---|---|---|---|
| best-of-n(refinement) | 3 | 5 | True | 33.78 | 33.78 | 100.0% | 50.0% |
| best-of-n(refinement) | 3 | 5 | False | 31.23 | 31.23 | 100.0% | 46.88% |
| best-of-n(refinement) | 5 | 3 | True | 31.85 | 31.85 | 100.0% | 50.0% |
| best-of-n(refinement) | 5 | 3 | False | 31.35 | 31.35 | 100.0% | 50.0% |
| refinement(best-of-n) | 3 | 5 | True | 32.66 | 51.32 | 63.64% | 48.48% |
| refinement(best-of-n) | 3 | 5 | False | 32.88 | 50.1 | 65.62% | 53.12% |
| **refinement(best-of-n)** | **5** | **3** | **True** | **34.88** | **57.56** | **60.61%** | **54.55%** |
| refinement(best-of-n) | 5 | 3 | False | 29.54 | 49.75 | 59.38% | 43.75% |
| best-of-n | N/A | 15 | True | 29.64 | 32.71 | 90.62% | 56.25% |
| best-of-n | N/A | 15 | False | 28.25 | 33.48 | 84.38% | 53.12% |

**Compfiles: Length Optimization** Consider Figure 1, a lemma from the 2022 IMO Question 2 (Compfiles) that we optimize for length. ImProver halves thr proof from 12 tactics to 6. Here, ImProver makes multiple nontrivial optimizations, such as eliminating the h2' and h4 and hxw hypotheses, as well as fully generating proof terms for specific rewrites and other tactics.

**Compfiles: Declarativity Optimization** Consider Figure 7, in which a lemma from the 2019 IMO problem 1 (from the Compfiles dataset) is optimized for declarativity. This introduces multiple new hypotheses, which generalize a linear_property of the functions, and then reuses and instantiates that (and others, too) hypothesis throughout the proof, creating a significantly more declarative proof.

**Original (human-written)**

```
lemma additive_to_int_linear (f : ℤ → ℤ) (h:
    ∀ (x y : ℤ), f (x + y) = f x + f y):
  ∃ c, ∀ a, f a = c * a  := by
 let g := AddMonoidHom.toIntLinearMap <|
   AddMonoidHom.mk' f h
 refine ⟨f 1, fun a => ?_⟩
 change g a = g 1 * a
 rw [mul_comm, ← smul_eq_mul, ←
   LinearMap.map_smul, smul_eq_mul, mul_one]
```

ImProver **(declarativity-optimized)**

```
lemma additive_to_int_linear (f : ℤ → ℤ) (h:
    ∀ (x y : ℤ), f (x + y) = f x + f y):
  ∃ c, ∀ a, f a = c * a   := by
 let g := AddMonoidHom.toIntLinearMap <|
   AddMonoidHom.mk' f h
 have linear_property : ∀ a, f a = g a := by
   intro a
   rfl
 have g_smul : ∀ a, g a = g 1 * a := by
   intro a
   rw [mul_comm, ← smul_eq_mul, ←
    LinearMap.map_smul, smul_eq_mul, mul_one]
 refine ⟨f 1, fun a => ?_⟩
 have f_eq_g : f a = g a := linear_property a
 have g_a_eq : g a = g 1 * a := g_smul a
 rw [f_eq_g, linear_property 1, g_a_eq]
```

Figure 7: Optimizing a lemma from IMO 2019 P1 for declarativity

**Compfiles: Mixed Optimization**    Consider Figure 8, in which a lemma from the 2023 USAMO problem 2 (from the Compfiles dataset) is optimized for mixed declarativity and length. This introduces a new hypothesis, which declares a powerful intermediate lemma, which is then applied to solve the problem. Moreover, this declarativity is introduced in such a way that it makes the proof more concise than the original, with 3 tactics rather than 5.

**Original (human-written)**

```
lemma lemma_3 {a b c : ℝ+} (h : a = b + c) :
    c < a  := by
  rw [h]
  obtain ⟨b, hb⟩ := b
  obtain ⟨c, hc⟩ := c
  rw [←Subtype.coe_lt_coe, Positive.coe_add]
  exact lt_add_of_pos_left c hb
```

ImProver **(mix-optimized)**

```
lemma lemma_3 {a b c : ℝ+} (h : a = b + c) :
    c < a    := by
  have : ↑c < ↑b + ↑c := lt_add_of_pos_left
    c.val b.property
  rw [h, ←Subtype.coe_lt_coe,
    Positive.coe_add]
  exact this
```

Figure 8: Optimizing a lemma from USAMO 2023 P2 for mixed declarativity/length

**MIL: Length Optimization**    Consider Figure 9, which optimizes an exercise solution from MIL Chapter 8, Section 1 (Group theory) for length, modifying the proof structure and introducing proof terms into the structure of the proof to shorten it from 9 tactic invocations to 7.

**Original (human-written)**

```
example (φ : G →* H) (ψ : H →* K) (S :
    Subgroup G) :
  map (ψ.comp φ) S = map ψ (S.map φ)  := by
  ext x
  simp only [mem_map]
  constructor
  · rintro ⟨y, y_in, hy⟩
    exact ⟨φ y, ⟨y, y_in, rfl⟩, hy⟩
  · rintro ⟨y, ⟨z, z_in, hz⟩, hy⟩
    use z, z_in
    calc ψ.comp φ z = ψ (φ z) := rfl
      _             = ψ y := by congr
```

ImProver **(length-optimized)**

```
example (φ : G →* H) (ψ : H →* K) (S :
    Subgroup G) :
  map (ψ.comp φ) S = map ψ (S.map φ)    :=
    by
  ext x
  simp only [mem_map]
  constructor
  rintro ⟨y, y_in, hy⟩; exact ⟨φ y, ⟨y, y_in,
    rfl⟩, hy⟩
  rintro ⟨y, ⟨z, z_in, hz⟩, hy⟩; exact ⟨z,
    z_in, (congr_arg ψ hz).trans hy⟩
```

Figure 9: Optimizing a lemma from the solutions of MIL CH08 S01 for length

**MIL Declarativity Optimization.**    Consider Figure 10, a result from MIL that we optimize for the declarative metric. This original proof carried a score of 0, as it has no `have` statements. In comparison, after applying ImProver, the model explicitly defines hypotheses for use in the latter half of the proof; these hypotheses can easily be converted into standalone lemmas for reuse.

**Original (human-written)**

```
def iso₁ [Fintype G] (h : Disjoint H K) (h' :
    card G = card H * card K)
: K ≃* G / H  := by
  apply MulEquiv.ofBijective
    ((QuotientGroup.mk' H).restrict K)
  rw [bijective_iff_injective_and_card]
  constructor
  · rw [← ker_eq_bot_iff, (QuotientGroup.mk'
    H).ker_restrict K]
    simp [h]
  · symm
    exact aux_card_eq h'
```

ImProver **(declarativity-optimized)**

```
def iso₁ [Fintype G] (h : Disjoint H K) (h' :
    card G = card H * card K)
: K ≃* G / H    := by
  have injectivity : Function.Injective
    ((QuotientGroup.mk' H).restrict K) := by
    rw [← ker_eq_bot_iff, (QuotientGroup.mk'
    H).ker_restrict K]
    simp [h]
  have card_eq : card (G / H) = card K := by
    exact aux_card_eq h'
  apply MulEquiv.ofBijective
    ((QuotientGroup.mk' H).restrict K)
  rw [bijective_iff_injective_and_card]
  constructor
  exact injectivity
  symm
  exact card_eq
```

Figure 10: Optimizing a group-theoretic result from MIL Chapter 8 Section 1 for declarativity.

**MIL: Mixed Optimization**  Consider Figure 11, which optimizes an exercise solution from MIL Chapter 8, Section 1 (Group theory) for mixed length/declarativity. ImProver significantly modifies the structure of the proof, applying more general-purpose tactics like `simpa` to simplify the proof (via eliminating the need for manual step-by-step calculations). Moreover, the proof introduces declarativity by a intermediate result, which is used by the `simpa` call to finish the proof in an efficient and declarative manner.

**Original (human-written)**

```
lemma eq_bot_iff_card {G : Type*} [Group G]
    {H : Subgroup G} [Fintype H] :
  H = ⊥ ↔ card H = 1  := by
 suffices (∀ x ∈ H, x = 1) ↔ ∃ x ∈ H, ∀ a ∈
    H, a = x by
   simpa [eq_bot_iff_forall, card_eq_one_iff]
 constructor
· intro h
  use 1, H.one_mem
· rintro ⟨y, -, hy'⟩ x hx
  calc x = y := hy' x hx
    _    = 1 := (hy' 1 H.one_mem).symm
```

**ImProver (mix-optimized)**

```
lemma eq_bot_iff_card {G : Type*} [Group G]
    {H : Subgroup G} [Fintype H] :
  H = ⊥ ↔ card H = 1     := by
 have : (∀ x ∈ H, x = 1) ↔ ∃ x ∈ H, ∀ a ∈
   H, a = x :=
   ⟨λ h => ⟨1, H.one_mem, h⟩, λ ⟨y, _, hy'⟩ x
   hx => (hy' 1 H.one_mem).symm ▷ hy' x hx⟩
 simpa [eq_bot_iff_forall, card_eq_one_iff]
   using this
```

Figure 11: Optimizing a lemma from MIL CH08 S01 solution for mixed declarativity/length

**Mathlib: Length Optimization**  Consider Figure 12, which optimizes a theorem in algebraic topology from mathlib for length, eliminating `simp` calls and combining tactics to shorten it from 3 tactic invocations to 1.

**Original (human-written)**

```
/-- If `f(p(t) = g(q(t))` for two paths `p`
    and `q`, then the induced path homotopy
    classes
`f(p)` and `g(p)` are the same as well,
    despite having a priori different types
    -/
theorem heq_path_of_eq_image : HEq ((π_m
    f).map ⟦p⟧) ((π_m g).map ⟦q⟧)  := by
 simp only [map_eq, ←
    Path.Homotopic.map_lift]; apply
    Path.Homotopic.hpath_hext; exact hfg
```

**ImProver (length-optimized)**

```
/-- If `f(p(t) = g(q(t))` for two paths `p`
    and `q`, then the induced path homotopy
    classes
`f(p)` and `g(p)` are the same as well,
    despite having a priori different types
    -/
theorem heq_path_of_eq_image : HEq ((π_m
    f).map ⟦p⟧) ((π_m g).map ⟦q⟧)    := by
  exact Path.Homotopic.hpath_hext hfg
```

Figure 12: Optimizing a theorem from `Mathlib/FundamentalGroupoid/InducedMaps` for length

**Mathlib: Declarativity Optimization**  Consider Figure 13, a theorem from Mathlib that we optimize for declarativity.

This original proof carried a score of 0, as it does not contain any declarative statements. It is concise and efficient, however, it is difficult to understand and read.

After optimizing for declarativity, we see that the model did not change the structure of the proof. Rather, it added an intermediate declaration so that users can better understand the state after the `convert`. This intermediate tactic greatly helps in the understandability and clarity of the proof.

**Mathlib: Mixed Optimization**  Consider Figure 14, a theorem from Mathlib that we optimize for mixed length/declarativity.

We observe that ImProver applies more complex tactics such as `all_goals` and `split_ifs` to significantly decrease the number of tactics in the proofs, while maintaining the overall structure.

**Original (human-written)**

```
/-- Another version of
    `simply_connected_iff_paths_homotopic` -/
theorem simply_connected_iff_paths_homotopic'
    {Y : Type*} [TopologicalSpace Y] :
  SimplyConnectedSpace Y ↔
    PathConnectedSpace Y ∧ ∀ {x y : Y} (p₁
  p₂ : Path x y), Path.Homotopic p₁ p₂  :=
  by
 convert
   simply_connected_iff_paths_homotopic (Y
   := Y)
 simp [Path.Homotopic.Quotient,
   Setoid.eq_top_iff]; rfl
```

ImProver **(declarativity-optimized)**

```
/-- Another version of
    `simply_connected_iff_paths_homotopic` -/
theorem simply_connected_iff_paths_homotopic'
    {Y : Type*} [TopologicalSpace Y] :
  SimplyConnectedSpace Y ↔
    PathConnectedSpace Y ∧ ∀ {x y : Y} (p₁
  p₂ : Path x y), Path.Homotopic p₁ p₂    :=
    by
 convert
   simply_connected_iff_paths_homotopic (Y
   := Y)
 have h1 : ∀ x y : Y, Subsingleton
   (Path.Homotopic.Quotient x y) ↔ ∀ {p₁ p₂
   : Path x y}, Path.Homotopic p₁ p₂ := by
   intro x y
   simp [Path.Homotopic.Quotient,
   Setoid.eq_top_iff]
   rfl
 simp only [h1]
```

Figure 13: Optimizing a theorem from `Mathlib/FundamentalGroupoid/SimplyConnected` for declarativity

**Original (human-written)**

```
theorem trans_refl_reparam (p : Path x₀ x₁) :
   p.trans (Path.refl x₁) =
     p.reparam (fun t => ⟨
   transReflReparamAux t,
   transReflReparamAux_mem_I t⟩) (by
   continuity)
       (Subtype.ext
   transReflReparamAux_zero) (Subtype.ext
   transReflReparamAux_one)  := by
    p.reparam (fun t => ⟨
   transReflReparamAux t,
   transReflReparamAux_mem_I t⟩) (by
   continuity)
       (Subtype.ext
   transReflReparamAux_zero) (Subtype.ext
   transReflReparamAux_one) := by
 ext
 unfold transReflReparamAux
 simp only [Path.trans_apply, not_le,
   coe_reparam, Function.comp_apply,
   one_div, Path.refl_apply]
 split_ifs
 · rfl
 · rfl
 · simp
 · simp
```

ImProver **(mix-optimized)**

```
theorem trans_refl_reparam (p : Path x₀ x₁) :
   p.trans (Path.refl x₁) =
     p.reparam (fun t => ⟨
   transReflReparamAux t,
   transReflReparamAux_mem_I t⟩) (by
   continuity)
       (Subtype.ext
   transReflReparamAux_zero) (Subtype.ext
   transReflReparamAux_one)    := by
    p.reparam (fun t => ⟨
   transReflReparamAux t,
   transReflReparamAux_mem_I t⟩) (by
   continuity)
       (Subtype.ext
   transReflReparamAux_zero) (Subtype.ext
   transReflReparamAux_one)  := by
 ext t
 simp only [Path.trans_apply,
   Path.refl_apply, coe_reparam,
   Function.comp_apply]
 unfold transReflReparamAux
 split_ifs
 all_goals { simp }
```

Figure 14: Optimizing a theorem from `Mathlib/FundamentalGroupoid/SimplyConnected` for mixed length/declarativity

**Full Proof Generation (MIL).** We analyze the application of ImProver to neural theorem proving in the MIL example from Figure 15. This theorem relating to group theory originally has no proof, however, ImProver generates one from scratch. This generated proof is verified to be correct by Lean, utilizing all the included hypotheses as well as a retrieved mathlib theorem.

**Original (human-written)**

```
example (φ : G →* H) (S T : Subgroup H) (hST
   : S ≤ T) : comap φ S ≤ comap φ T := by
 sorry
```

ImProver **(completeness-optimized)**

```
example (φ : G →* H) (S T : Subgroup H) (hST
   : S ≤ T) : comap φ S ≤ comap φ T   := by
 intro g
 simp only [mem_comap]
 intro hS
 exact hST hS
```

Figure 15: Solving a group theorem exercise from MIL Chapter 8 Section 1.

**Full Proof Generation (MiniF2F).** We analyze the application of ImProver to neural theorem proving in the MiniF2F example from Figure 16. This theorem relating to number theory originally

has no proof, however, ImProver generates one from scratch by calculating step by step, applying theorems retrieved from Mathlib, as well as general tactics like `simp`.

**Original (human-written)**

```
theorem mathd_numbertheory_345 :
  (2000 + 2001 + 2002 + 2003 + 2004 + 2005 +
    2006) % 7 = 0 := by sorry
```

ImProver **(completeness-optimized)**

```
theorem mathd_numbertheory_345 :
  (2000 + 2001 + 2002 + 2003 + 2004 + 2005 +
    2006) % 7 = 0    := by
calc
  (2000 + 2001 + 2002 + 2003 + 2004 + 2005 +
    2006) % 7 = (0 + 1 + 2 + 3 + 4 + 5 + 6)
    % 7 := by
    simp [Nat.add_mod,
  Nat.mod_eq_zero_of_dvd, Nat.dvd_refl,
  Nat.mod_add_mod, Nat.mod_self]
  _ = 21 % 7 := by simp
  _ = 0 := by simp
```

Figure 16: Solving a number theoretic theorem from MiniF2F.

