# OpenReview forum: "ImProver: Agent-Based Automated Proof Optimization"
_ICLR.cc/2025/Conference — ICLR 2025 Poster_

### Official Review · Reviewer_gxhK · 2024-10-31

**Soundness:** 3
**Presentation:** 3
**Contribution:** 2
**Rating:** 8
**Confidence:** 3

**Summary:**

The paper proposes ImProver, a framework designed to optimize proofs based on various metrics, such as shortening proof length and enhancing human readability. The authors establish several performance metrics to evaluate the quality of improvements and conduct comprehensive case studies across representative benchmark datasets for an extensive analysis.

**Strengths:**

- As the first work in proof optimization, this paper defines novel and reasonable metrics to quantify the quality of Lean proofs.
- The paper integrates several advancements from the LLM community, including proof generation, retrieval, chain of thought (CoT) reasoning, and self-correction in code generation, for optimizing proofs. The inclusion of these methods is well-motivated, and the experiments are extensive. Furthermore, ImProver achieves notable performance without using models fine-tuned specifically for proof generation, showing its significant contributions to proof synthesis and optimization.
- The open-source code is well-structured and documented, facilitating easy adoption and use by the community.

**Weaknesses:**

While the experimental results are comprehensive, their presentation could be improved, which is particularly important for a paper focused on advancing LLM applications in proof optimization. Specifically, in Section 4.2.1 on ablation testing:

- At line 450, the term “string list” is introduced without prior definition. Could the authors clarify the relationship between "string list" and the "flat" format mentioned earlier? This would help ensure consistency in terminology throughout the paper.
- Table 5 could be clarified further. Upon cross-checking, it appears that the best/worst values are based on configurations giving the highest “improvement” scores, whereas the last row presents the best scores across all configurations. It would be helpful to explicitly state this distinction in the main text to enhance clarity.

**Questions:**

- Could the authors elaborate further on the “flat” and “structured” formats described in Section 3.1.2 and provide examples for clarity?
- When selecting parameters in the ablation study, why was the best parameter combination chosen based on improvement rather than accuracy? Could the authors discuss the trade-offs between optimizing for improvement versus accuracy, and how this choice might impact the practical applicability of ImProver?

---

> ### Author Response · Authors · 2024-11-26
>
> Thank you for your review and your feedback on our paper. We have taken this into account to improve the quality and rigor of our work. Attached below is a point-by-point response to your comments and questions.
>
> ### 1. Terminology and Presentation
> > Could the authors clarify the relationship between "string list" and the "flat" format mentioned earlier?
>
> > Could the authors elaborate further on the “flat” and “structured” formats described in Section 3.1.2 and provide examples for clarity?
>
> We have clarified terms like “string list” and “flat format” to ensure consistent usage of such terms throughout the paper. Namely, a “flat” format is equivalent to outputting a “string list,” and a “structured” format is equivalent to a “string tree”. For example, a "string list" would be of the general form `["tactic 1", "tactic 2", ...]`, whereas a "string tree" would be of the general form:
> ```
> [("tactic 1", [
> 	("tactic 1a",[...]),
> 	("tactic 1b",[...]),
> 	...]
>  ),
>  ("tactic 2", [...])
> ...]
> ```
>
>
>
>
> ### 2. Experimental Clarity
>
> > It would be helpful to explicitly state this distinction in the main text to enhance clarity.
>
> > When selecting parameters in the ablation study, why was the best parameter combination chosen based on improvement rather than accuracy? [...]
>
> The ablation process and the “improvement”-based selection have been explicitly justified and clarified in our revision (section 4.1.1) to account for the reviewer’s feedback and ensure a clear presentation of our experimental process.
>
> Specifically, we justify the use of the improvement score for the selection of the “best” parameter combination or model, as this improvement score represents the expected improvement in the metric score – accounting for possible errors in the generation. This is preferable to the other possible performance metrics, such as accuracy, as it prioritizes both correctness, as well as large improvements in metric scores. Indeed, improvement penalizes incorrect generations and adds additional weight to greater improvements in the metric score, whereas accuracy and nonzero accuracy solely describe the correctness of generations. This is useful in benchmarking the improvements of generation abilities between the base model and ImProver. Conversely, nonempty improvement ignores incorrect submissions, describing the ability of the model to improve a given theorem, assuming correct generations. Improvement amalgamates all of this, rewarding correct and significant improvements and discouraging incorrect generations.

---

> > ### Comment · Reviewer_gxhK · 2024-11-29
> >
> > I thank the authors for their detailed responses, as well as the added experiments. All my concerns has been addressed, and with the more comprehensive insights provided by additional experiments, I have increased my rating to 8 (accept).

---

> > > ### Author Response · Authors · 2024-12-03
> > >
> > > We greatly appreciate your consideration of our work and revisions, as well as your patience and thoughtful comments throughout this discussion period. Thank you again for your insights and advice to help improve the quality and robustness of our work!

---

### Official Review · Reviewer_3uHZ · 2024-11-01

**Soundness:** 4
**Presentation:** 4
**Contribution:** 4
**Rating:** 8
**Confidence:** 4

**Summary:**

Authors are studying whether LLMs can optimize formal math proofs for additional criteria besides correctness, such as length and readability. They find that naively applying LLMs don’t perform great at this task, but that multiple interventions like augmenting the formal proof with comments, adding relevant context via retrieval, or selecting the best of N solutions, substantially improves performance. Finally, they ablate these interventions and find that each contributes substantially to the overall performance.

**Strengths:**

The authors frame an interesting and potentially very impactful problem, propose a smart solution, and execute on that solution well. The paper is well-written and the results appear convincing to me.

**Weaknesses:**

The authors could mention interactive proofs / prover-verifier games (https://arxiv.org/abs/2108.12099), where LLMs are trained to write informal math proofs to be maximally legible to an independent verifier, as related work.

I believe the caption of Figure 1 should read (left) instead of (right)?

For reproducibility, the authors should mention the exact version of GPT-4o they used for their experiments (gpt-4o-2024-08-06 ?)

I’m taking it on trust that “readability” defined as in the paper is a sensible definition - to someone not familiar with Lean, the "readability-optimized" version of the proof in Figure 3 is not actually a lot easier to parse than the other version.

**Questions:**

Idle musing, feel free to ignore: Do you have thoughts about the ceiling/floor of your proposed metrics - are there degenerate solutions? In particular, could `we evaluate this using the ratio of number of explicitly typed have tactics to total number of tactic invocations.` incentivize the model to generate a lot of superfluous `have` tactics to inflate this metric?

---

> ### Author Response · Authors · 2024-11-26
>
> Thank you for your suggestions, comments, and feedback. We deeply appreciate your review and have attached below a full response to your questions and feedback.
>
> ### 1. Related Work
> > The authors could mention interactive proofs/prover-verifier games
>
> As per the reviewer’s suggestion, we have added a greater elaboration in the Related Work section, including a discussion on interactive proofs and prover-verifier games, noting that Lean constitutes a sound verifier in the context of Interactive Proof Systems.
>
> ### 2. Figure
> >I believe the caption of Figure 1 should read (left) instead of (right)?
>
> We have revised the caption to fix the error and any other obfuscations.
>
>
>
> ### 3. GPT Version
> >For reproducibility, the authors should mention the exact version of GPT-4o they used for their experiments (gpt-4o-2024-08-06 ?)
>
> In our revision, we have now included the GPT version explicitly in our initial description of ImProver (Section 4.1).
>
>
>
> ### 4. Readability and Degenerate Solutions
> > I’m taking it on trust that “readability” defined as in the paper is a sensible definition
>
> We appreciate the feedback on the intuitiveness of our readability metric, as the concept of what constitutes a "readable" proof is quite complex and isn't so simple as to be defined as a simple ratio of `have` tactics. Indeed, when initially defining readability in section 3.1, we wrote that readability is defined as the declarativity of the proof -- a notion that doesn't perfectly align with the intuition of what constitutes a readable proof.
>
> As such, to ensure clarity on this point, we have renamed the “readability metric” to the “declarativity metric”, which has a more objective definition as per [1, 2] that is well-modeled by the ratio-based scoring function. We’ve additionally included side-by-side examples of proofs written in a declarative and non-declarative style to qualitatively describe the metric (these examples are a subset of the examples forwarded to the model itself). We justify this change by the definition of the readability metric , and note that as the model prompt [Appendix A.2] itself is unchanged by this renaming, this change is purely syntactic, and has no effect on the model behavior.
>
> > [...] Are there degenerate solutions?
>
> Your concern for degenerate solutions is quite valid.  We acknowledge that the metric is simplistic and subject to potential gaming (e.g., through excessive use of have tactics). However, it serves as a starting point for exploring proof optimization capabilities. We have added clarifications that the tested metrics are simplistic in definition and designed to evaluate the model’s adaptability and performance, with the intention of extending this work to more nuanced metrics.
>
> Moreover, in our revised submission, we explicitly discuss how examples guide the model in avoiding trivial solutions, such as the superfluous `have` tactics you mentioned. Although this empirically generates data that is free of these degenerate solutions, there is an underlying issue of scalability with this multi-shot based methodology, as for more complex metrics, large example sets become harder to construct and the possible edge cases where a degenerate solution may occur increases. To account for this, we discuss incorporating LLM-based scoring into metrics, which is natively supported in the ImProver framework.
>
>
> ## References:
> [1] Serge Autexier and Dominik Dietrich. A tactic language for declarative proofs. In Matt Kaufmann and Lawrence C. Paulson (eds.), Interactive Theorem Proving, pp. 99–114, Berlin, Heidelberg, 2010. Springer Berlin Heidelberg.
>
> [2] Freek Wiedijk. Formal proof sketches. In Stefano Berardi, Mario Coppo, and Ferruccio Damiani (eds.), Types for Proofs and Programs, pp. 378–393, Berlin, Heidelberg, 2004. Springer Berlin Heidelberg. ISBN 978-3-540-24849-1.

---

> > ### Comment · Reviewer_3uHZ · 2024-11-26
> >
> > Thank you for your response, I'm adjusting my confidence upwards!

---

> > > ### Author Response · Authors · 2024-12-03
> > >
> > > We sincerely thank you for your thoughtful feedback and comments throughout this discussion period, and are grateful for your consideration of our work and revisions these last few weeks!

---

### Official Review · Reviewer_XEdk · 2024-11-04

**Soundness:** 3
**Presentation:** 2
**Contribution:** 2
**Rating:** 5
**Confidence:** 3

**Summary:**

The paper tackles the proof optimization problem, which aims to optimize partly written proofs with respect to a given criterion (such as length and readability).
The paper also addresses the problem by querying to LLMs and ranking the outputs with the criterion.

**Strengths:**

+ The paper tackles the proof optimization, which has not been considered yet in neural theorem proving, and proposes a (simple) approach to it.
+ The proposed approach can outperform the baseline, which simply queries to an LLM (GPT-4o).

**Weaknesses:**

- Because the proposed approach to the proof optimization (described in Section 3) is simple, I find the main contribution of the paper is in applying LLMs to proof optimization, a problem in theorem proving. When evaluating the paper from such a viewpoint, I admit the proof optimization problem may be worth studying, but I don't think the paper argues its value in a convincing manner. The second paragraph of Section 1 discusses the importance of the proof optimization, but there is no reference to support the argument. That is, there is no objective material to find the importance of the problem. Therefore, it is unclear and unconvincing what benefit is expected by solving the proof optimization problem.

- Some experimental setting and terminology are unclear (see the following questions).

Typos:

- l35: precision --> imprecision

**Questions:**

- It is possible to explain the benefits of solving the proof optimization problem with application domains (such as formal verification or mathematical education), references, and concrete examples that support the claim?
- I think it is important to incorporate some kind of random in using the best-of-n technique. Does the proposed approach ensure diversity in the LLM outputs when using the best-of-n technique? If yes, how it is achieved? Otherwise, how do you address the potential issue of deterministic outputs?
- Can you provide a detailed description of the input format for ImProver and GPT-4o in the experiment? Especially, does the input include the original proof from the dataset, a partial proof, or start with an empty proof?
- The paper claims ImProver can rewrite proofs so that they are more modular. Can you provide a clear definition of 'modular' in the paper?  It would also be helpful to give modular and non-modular examples.
- What does the paper mean by efficient proofs? (E.g., it says "These proofs are extremely efficient" in lines 266-267).

---

> ### Author Response · Authors · 2024-11-26
>
> We greatly appreciate your feedback and comments on our submission. We have made revisions based on this feedback, and have included below a response to each of your comments and questions.
>
> ### 1. Importance of Proof Optimization
> > [...] Therefore, it is unclear and unconvincing what benefit is expected by solving the proof optimization problem.
>
> We have expanded the introduction to provide a more compelling argument for the value of proof optimization, supported by explicit references and applications to education, research, and training machine learning models.
>
> Specifically, we discuss the utility of proof optimization for increasing the understanding and decreasing external dependencies for better maintainability [1] for large libraries like Mathlib, as well as creating well-structured datasets for training ML models. Additionally, since the original submission, ImProver has shown to be invaluable for both these applications – having already made contributions to public Lean datasets (references removed for anonymity, but will be included in a final camera-ready version) – as well as currently being used to fine-tune and apply policy optimization for building a powerful neural theorem proving model.
>
> ### 2. Diversity in Best-of-N Outputs
> > I think it is important to incorporate some kind of random in using the best-of-n technique.
>
> In our revision, we clarify in the experimental setup (Section 3.2.3) that temperature settings are used to ensure diverse outputs during the best-of-N selection process.
>
>
>
> ### 3. Input Format
> > Can you provide a detailed description of the input format for ImProver and GPT-4o in the experiment?
>
> We have added a detailed description of the input formats for ImProver and GPT-4o, including examples to appendix A.1 and section 4.1. Note that the original proof is always included in the input prompt, as well as all relevant contextual information – although the ImProver prompt is heavily augmented with additional information such as dependencies, RAG, and chain-of-states.
>
>
>
> ### 4. Modular Proofs
> > Can you provide a clear definition of 'modular' in the paper?
>
> We acknowledge that the “modularity” of a proof was not adequately defined, as it was intended as an alias for readability and declarativity. We have revised this to consistently describe the notion as “declarative” throughout the submission, as well as properly describe and give side-by-side comparisons of declarative and nondeclarative proofs [Appendix A.3].
>
>
>
> ### 5. Efficient Proofs
> > What does the paper mean by efficient proofs?
>
> We acknowledge that the notion of “efficient proofs” was subjective and poorly defined, and was moreover redundant to the description of the Mathlib dataset as having concise and generalized proofs. The claim of “efficient” proofs has since been removed, now simply describing the proofs within Mathlib as being concise and generalized with many external dependencies.
>
>
> ## References:
> [1] https://leanprover-community.github.io/contribute/index.html

---

> > ### Comment · Reviewer_XEdk · 2024-11-30
> >
> > Thank you for the response.
> >
> > Even after reading it, I am concerned about the formalization of proof optimization in the current form. A clear application of proof optimization (for me) is the maintenance of mathematical libraries (the other applications described in the introduction would be possible but seem still potential), but it is unclear whether the style guidelines of proofs mentioned in the revision can be represented by a metric of the proposed approach (as pointed out by Reviewer 5mDw, formalizing the guidelines as such a metric looks challenging). If the paper showed the metric implementation of the guidelines and the proposed approach actually improves the proofs according to it, the contributions of the paper would be significantly enhanced.
> >
> > That said, the revision is more well motivated than the original submission, the concerns described above may be a problem in a next step, and the response resolves the other concerns in my initial review. Therefore, I don't disagree with acceptance of the paper.
> >
> > Minor comments:
> > - l047: "to ensure efficient and generalized theorems" This would be misleading because I don't think the style guidelines to ensure generalized theorems are given as a metric because the metric should be on proofs, not theorems.
> > - l600-602: "The mathlib Community. ... POPL'20" --> CPP'20

---

> > > ### Author Response · Authors · 2024-12-03
> > >
> > > Thank you for your response and consideration of our revisions and work throughout this discussion period; your comments and suggestions have been invaluable to improving the clarity and quality of our work.
> > >
> > > With regards to your above comments, we have modified l047 to describe the Mathlib style guidelines as prioritizing “efficient and interdependent proofs”, as the prior description was intended to describe the high level of dependencies (inaccurately described as “generalized”) and concise proofs used throughout the Mathlib corpus, which was inconsistent and unclear with our work’s focus on proofs rather than theorems and their statements. Additionally, the mistake in l600-602 has been corrected. We hope to showcase these changes in a final camera-ready copy.
> > >
> > > Additionally, we acknowledge the difficulty of formalizing the notion of the “quality” of a proof, and note that this difficulty is mitigated by designing more explicit metrics based on the use-case. For example, our old “readability” metric was quite unclear and subjective notion, but reshaping the metric to “declarativity”, which is explicitly described in [1,2], is far more faithful as the quantitative measurements of improvement better coincides with an observed improvement in the level of declarative statements throughout a given proof. Although the metrics we analyze here are intentionally simplistic in comparison to Mathlib style guidelines in their entirety, in future versions, we hope to approximate corpus style guidelines via the combination of many smaller explicit metrics – as this would greatly benefit the work of formalizers and maintainers alike.
> > >
> > > We greatly appreciate the time and effort you have spent throughout this discussion period to review and offer your insights and knowledge to help us improve this work, and we hope to fully incorporate these insights in a final camera-ready version soon. Thank you again for your patience and coaching, and we look forward to expanding our work further in the future!
> > >
> > >
> > >
> > > [1] Serge Autexier and Dominik Dietrich. A tactic language for declarative proofs. In Matt Kaufmann and Lawrence C. Paulson (eds.), Interactive Theorem Proving, pp. 99–114, Berlin, Heidelberg, 2010. Springer Berlin Heidelberg.
> > >
> > > [2] Freek Wiedijk. Formal proof sketches. In Stefano Berardi, Mario Coppo, and Ferruccio Damiani (eds.), Types for Proofs and Programs, pp. 378–393, Berlin, Heidelberg, 2004. Springer Berlin Heidelberg. ISBN 978-3-540-24849-1.

---

### Official Review · Reviewer_5mDw · 2024-11-04

**Soundness:** 3
**Presentation:** 4
**Contribution:** 3
**Rating:** 6
**Confidence:** 4

**Summary:**

The paper introduces ImProver, a large language model (LLM) agent designed for automated proof optimization in the proof assistant Lean. ImProver adopts several LLM augmentation methods, including a Chain-of-States technique that provides intermediate proof states to LLM, retrieval methods, and sampling methods.

The paper defines three key metrics for evaluation: the Length Metric, which aims to minimize the number of tactic invocations; the Readability Metric, which favors a declarative haveproof style; and the Completion Metric, which measures proof correctness.

In experiments, the author compares ImProver with the baseline GPT-4o model on Mathematics in Lean (MIL), Compfiles, and Mathlib datasets. The results show that ImProver reduces the length of 35.44% of proofs compared to 8.31% for GPT-4o, and increases readability in 24.56% of proofs compared to 6.13% for GPT-4o. Proof completion was evaluated only on the selected MIL dataset, where ImProver achieved 39.13% successful proof generation compared to 21.73% for GPT-4o.

**Strengths:**

This paper first discovers a new field of automated proof optimization, and conducts a pioneering attempt in this field, marking the first significant effort to use LLMs for enhancing formal proofs with respect to certain metrics in proof assistants like Lean. A notable strength of this work is its emphasis on the role of proof states in guiding LLMs. By introducing the Chain-of-States technique, which provides intermediate proof states to the LLM, the authors demonstrate an understanding of how contextual information can significantly improve the accuracy and efficiency of proof optimization.

**Weaknesses:**

- The method proposed in this paper is well-suited for metrics that can be directly computed from the formal proof. However, this limits its applicability to more complex metrics such as readability. The paper defines readability as the ratio of explicitly typed have tactics to the total number of tactic invocations in line 130. This simple definition may lead to a proof with numerous unused have tactics being considered more readable than a normal proof. The reviewer considers readability to be a multifaceted metric, influenced by various factors including the use of underlines, the depth of proof terms, the naming of variables, appropriate comments within the proof, and other stylistic elements. These factors are challenging to quantify with a simple function, suggesting that the proposed method may not be effective for assessing readability or other nuanced aspects of proof quality.
- The reviewer suggests developing a combined metric that considers both length and readability for a more accurate assessment of proof quality in real applications, since improving in one metric may significantly sacrificing other metrics. Additional experiments should test whether the framework described in the paper can effectively balance and improve these combined metrics.
- The proof generation results in Table 7 are limited to specific selected datasets. The reviewer believes that ImProver's neural theorem proving results should also be evaluated on unselected datasets and more commonly used test sets, and compared with a wider range of models to provide a comprehensive assessment of the proposed method's performance.
- The comparison with GPT-4o, which lacks proper Lean 4 syntax capabilities, may not accurately reflect the effect of RAG or refinement policies of ImProver. More ablation studies could be conducted to demonstrate that the rise in performance is not solely due to the guidance for correct Lean 4 syntax in the prompt.
- The reviewer suggests that rule-based optimization methods for Lean, particularly for the length metric (e.g., using nested termwise proofs to replace tactic-based proofs), exist. The authors should compare ImProver with these methods to demonstrate its relative strengths.

**Questions:**

- Could the authors clarify under what criterion the mathematical branch of the dataset for testing the neural theorem proving effect was selected and whether they have evaluated the method on the entire dataset and other commonly used datasets?
- Could the author clarify whether they have tried any rule-based methods and compared their results to those produced by ImProver?

---

> ### Author Response · Authors · 2024-11-26
>
> **(1/2)** Thank you for your in-depth feedback and review of our work. It has been incredibly helpful in improving the robustness and clarity of our paper. Below is a point-by-point response to each of your questions and comments.
>
> ### 1a. Readability Metric:
> > The reviewer considers readability to be a multifaceted metric, influenced by various factors including the use of underlines, the depth of proof terms, the naming of variables, appropriate comments within the proof, and other stylistic elements. These factors are challenging to quantify with a simple function, suggesting that the proposed method may not be effective for assessing readability or other nuanced aspects of proof quality.
>
> We appreciate the reviewer's concern about the complexity of defining readability, and acknowledge that defining a readability metric in its full generality is more complex than the ratio of “have” tactics, involving many subjective and stylistic factors that may be difficult to quantify. Indeed, the goal of many formalizers is to generate “efficient” or “elegant” proofs – metrics that are exceedingly difficult to define and quantify, and therefore difficult to optimize for with a simple function.
>
> However, as building good metrics is left as a user-level task, we take advantage of the fact that in applications where proofs need to be optimized to some specific standard, a general measure of a “readable” or “efficient” proof is not necessary, as we simply need to conform to some more strictly defined notion of what the optimal proof style is. For example, in training ML models on Lean data, it is valuable to explicitly construct the proof argument and structure in a declarative manner, and in making large libraries like Mathlib, it is valuable to have proofs be as concise as possible and use dependencies (lemmas) from the library rather than internal subproofs (haves).
>
> With this in mind, we acknowledge that the metric is simplistic and subject to potential gaming (e.g., through excessive use of have tactics). However, it serves as a starting point for exploring proof optimization capabilities. We have added clarifications that the tested metrics are simplistic in definition and designed to evaluate the model’s adaptability and performance, with the intention of extending this work to more nuanced metrics. Additionally, we are currently testing an additional metric based on the number of external dependencies of an inputted proof, which we hope to include in the next revision by the end of the discussion period to better highlight the proof optimization capabilities of ImProver over arbitrary metrics.
>
> Moreover, to ensure clarity on this point, we have renamed the “readability metric” to the “declarativity metric”, which has a more objective definition as per [1, 2] that is well-modeled by the ratio-based scoring function. We’ve additionally included side-by-side examples of proofs written in a declarative and non-declarative style to qualitatively describe the metric (these examples are a subset of the examples forwarded to the model itself). We justify this change by the definition of the readability metric , and note that as the model prompt [Appendix A.2] itself is unchanged by this renaming, this change is purely syntactic, and has no effect on the model behavior.
>
>
>
> Additionally, we have further emphasized the flexibility of our approach, empowering users to define their own metrics and incorporating LLM-based scoring. This facilitates experimentation with complex, human-intuitive definitions of readability.
>
>
> ### 1b. Degenerate Solutions:
>
> > The paper defines readability as the ratio of explicitly typed have tactics to the total number of tactic invocations in line 130. This simple definition may lead to a proof with numerous unused have tactics being considered more readable than a normal proof.
>
> The potential for degenerate solutions is a valid concern. In our revised submission, we explicitly discuss how examples guide the model in avoiding trivial solutions, such as generating excessive have tactics in the declarativity metric. Although this empirically generates data that is free of these degenerate solutions, there is an underlying issue of scalability with this multi-shot based methodology, as for more complex metrics, large example sets become harder to construct and the possible edge cases where a degenerate solution may occur increases. To account for this, we discuss incorporating LLM-based scoring into metrics, which is natively supported in the ImProver framework.

---

> > ### Author Response · Authors · 2024-11-26
> >
> > **(2/2)**
> > ### 2. Combined Metric:
> > > The reviewer suggests developing a combined metric that considers both length and readability for a more accurate assessment of proof quality in real applications [...]
> >
> > We concur with the reviewer’s suggestion of developing more complex, combined metrics that better describe a sense of “proof quality” in real-world applications.
> >
> > We plan to implement this and showcase the generality of our approach by implementing a dependency metric. Inspired by feedback from expert formalizers, the overall quality of a proof was framed with respect to its dependencies: depending on the application, a “good” proof was one that either maximized the number of external dependencies, thereby minimizing redundant steps and using lemmas as much as possible, and in practice, shortening the proof a good deal. Or alternatively, minimized the number of external dependencies, constructing a self-standing theorem that builds from first principles (oftentimes resulting in many subproofs and a larger length). The former tends to be preferable in corpus-building and research, and the latter for pedagogy and model training.
> >
> > We are currently in the process of building such a metric and hope to provide experimental results by the end of the discussion period. We do re-emphasize that our focus here was on setting up the general Improver framework, which can then motivate the development of more nuanced metrics in the future.
> >
> > ### 3. Evaluation on Additional Datasets
> > > The reviewer believes that ImProver's neural theorem proving results should also be evaluated on unselected datasets and more commonly used test sets [...]
> >
> > We are currently testing ImProver’s neural theorem proving abilities on the miniCTX dataset, which includes theorems from real-world projects – similar to the actual use cases of ImProver. We hope to include these results in the next revision by the end of the discussion period.
> >
> >
> > ### 4. Clarification of GPT-4o Comparison
> > > More ablation studies could be conducted to demonstrate that the rise in performance is not solely due to the guidance for correct Lean 4 syntax in the prompt.
> >
> > We agree that the comparison with GPT-4o may conflate Lean syntax guidance with model performance. To address this, we have added an ablation study isolating the impact of syntax-specific prompting on performance by not forwarding error information to subsequent refinement stages. This demonstrates that improvements are not solely attributable to syntax guidance.
> >
> >
> > ### 5. Rule-Based Optimization Comparison
> > > The reviewer suggests that rule-based optimization methods for Lean, particularly for the length metric (e.g., using nested termwise proofs to replace tactic-based proofs), exist.
> >
> > We acknowledge that rule-based and symbolic methods of length optimization or other simple metrics exist, and with certain (simple) metrics, these symbolic methods can far outperform ImProver. Namely, with the length metric, it is possible to convert any tactic proof into a single-line proof term, and use this with the “exact” tactic to ensure that all proofs have length 1.
> >
> > However, this proof-term conversion strategy can be seen as a degenerate solution to the length metric, and as mentioned in part (1a) and (1b), a way to mitigate the existence of degenerate solutions is to utilize a reward-model rather than a reward function. Moreover, the rule-based methods exist to solely optimize a specific metric, and are not at all flexible to arbitrary metrics - or even exist for certain metrics at all. As such, for more complex metrics, such rule-based methods are not flexible enough to be usable at scale.
> >
> > As the proof-term conversion strategy generates degenerate solutions which are undesirable, ImProver allows for proof length optimization on a structural level that does not solely employ proof-term conversion like the rule-based methods do.
> >
> >
> >
> >
> >
> > ## Questions:
> > 1. The dataset for testing neural theorem proving was selected due to its relatively low complexity (undergraduate level) and relatively low -- but nonzero --number of external dependencies (as compared to research datasets and corpora like Mathlib, or self-standing competition problems like Compfiles). However, as discussed in (3), we concur that having more rigorous testing done on a more mainstream dataset would be more desirable, and as such, we are currently testing on the miniCTX dataset, whose results we hope to release in the next revision.
> > 2. See (5).
> >
> >
> >
> > ## References:
> > [1] Serge Autexier and Dominik Dietrich. A tactic language for declarative proofs. In Matt Kaufmann and Lawrence C. Paulson (eds.), Interactive Theorem Proving, pp. 99–114, Berlin, Heidelberg, 2010. Springer Berlin Heidelberg.
> >
> > [2] Freek Wiedijk. Formal proof sketches. In Stefano Berardi, Mario Coppo, and Ferruccio Damiani (eds.), Types for Proofs and Programs, pp. 378–393, Berlin, Heidelberg, 2004. Springer Berlin Heidelberg. ISBN 978-3-540-24849-1.

---

> > > ### Author Response · Authors · 2024-11-28
> > >
> > > ### 2. Combined Metric
> > >
> > > We have revised our original plan to implement a dependency metric in favor of your suggestion for a combined length/declarativity metric. This metric is now described in section 3.1, and results in 4.1, and additional details in appendix A.
> > >
> > > ### 3. Evaluation on Additional Datasets
> > >
> > > We have evaluated ImProver on the MiniF2F-test dataset at 8 samples. This is described in appendix B.1 and analyzed in section 4.2.2

---

> > > > ### Comment · Reviewer_5mDw · 2024-12-01
> > > > **Official comment by reviewer 5mDw**
> > > >
> > > > Dear Author(s),
> > > >
> > > > Thank you for your detailed response. It clarifies most of my concerns. I have changed the soundness from 2 to 3 and the rating from 5 to 6. For more specific comments, please see below.
> > > >
> > > > 1. The reviewer appreciates the additional experiments on combined metrics. This shows stronger support for the claim that the pipeline proposed in this paper is capable of improving with respect to complex user-defined metrics.
> > > >
> > > > 2. The reviewer believes that the name "declarativity" is more faithful than the original name "readability." The paper’s soundness has improved significantly after the renaming and the addition of paragraphs discussing the avoidance of possible degeneracies. Please note that there is a minor typo (an unchanged "readability") in Table 1.
> > > >
> > > > 3. The reviewer acknowledges this paper’s contribution in proposing a flexible pipeline for improving proofs with respect to arbitrary user-defined numerical metrics. However, the reviewer also cares about its practical performance and comparison with previous works. In particular, as mentioned in Section 2, many previous works on neural theorem proving in interactive theorem provers exist. Currently, there is no comparison with the results of these works, only a comparison with the baseline GPT-4 results.

---

> > > > > ### Author Response · Authors · 2024-12-03
> > > > >
> > > > > We thank you for your thoughtful advice, suggestions, and feedback throughout this discussion period; your insights helped us construct a far more robust and comprehensive work, as well as guiding our future work on proof optimization.
> > > > >
> > > > > Additionally, we thank you for your consideration of our revisions, and in regards to your specific comments, we have fixed all such typos for a final camera-ready copy. Moreover, we acknowledge our limited comparison in neural theorem proving methods, as the miniF2F comparison made in the revised Table 8 covers ImProver, GPT-4o, and the outperformant Lean Expert Iteration [1]. This comparison could be made more comprehensive, as we initially intended for this NTP comparison to simply show empirically that proof optimization is indeed an extension of neural theorem proving, but a more robust comparison would serve to better understand the practical performance of ImProver on NTP tasks. We hope to expand the comparison to additional NTP methods in a final camera-ready version.
> > > > >
> > > > > We are truly grateful for the valuable time you invested into this discussion period, and for your insights and advice on improving and developing our work further.
> > > > >
> > > > >
> > > > >
> > > > >
> > > > >
> > > > >
> > > > >
> > > > > [1] Wu, Z., Huang, S., Zhou, Z., Ying, H., Wang, J., Lin, D., & Chen, K. (2024). InternLM2. 5-StepProver: Advancing Automated Theorem Proving via Expert Iteration on Large-Scale LEAN Problems. arXiv preprint arXiv:2410.15700.

---

### Author Response · Authors · 2024-11-26

We sincerely thank all reviewers for their thoughtful and constructive feedback, which has provided valuable insights for improving the clarity, robustness, and depth of our work. We appreciate the reviewers’ recognition of the utility and importance of automated proof optimization, and our contributions to this problem via ImProver.

With this feedback, we revised our paper to better account for the reviewers’ critiques. We intend for this revision to be the first of two revisions, the latter of which will finalize and present the results of our ongoing experiments. Additional rebuttals, responses, and answers to questions will be sent to individual reviewers separately below.

We first summarize the core contribution and motivation behind ImProver. With recent advances in the development and popularization of interactive theorem provers such as Lean, formal mathematical research has been empowered to work at scale, with large teams collaborating on research [3] and corpora-building [1] in a manner not seen before in traditional mathematics research. Additionally, the existence of a perfect verifier has led to an interesting machine-learning opportunity to build a model that can construct theorems and proofs automatically. However, in both these cases, there are many instances where multiple proofs of a statement may all be valid, but there are specific criteria in which one is preferable to another.

To this end, we define and study the problem of automated proof optimization, in which we aim to generate a new proof based on a template proof that is both correct and optimized for an arbitrary, user-defined metric. To mathematicians, the ability to improve proofs automatically is invaluable to the maintenance and development of libraries for research [1,3] and pedagogy [2] alike. Additionally, in machine learning applications, proof optimization is a form of data augmentation: the limited amount of formal training data is currently a bottleneck for machine learning, and our methods provide ways of generating additional data automatically. More interestingly, our methods also provide a means of optimizing training data, as other work [4] suggests that generating formal proofs by “sketching” a high-level outline of a proof, which is then filled in by symbolic automated reasoning methods, is a promising methodology. For that purpose, our methods provide a means of generating such structured proofs from less structured ones.

We present ImProver, an initial approach that applies language model agents to the automated proof optimization problem. ImProver can rewrite formal proofs in Lean to optimize for arbitrary user-defined metrics, such as length or declarativity, and perform generalized neural theorem proving. We evaluate ImProver on proof optimization tasks over real-world datasets at varying levels of complexity, in which it outperforms a GPT-4o baseline, as well as perform ablations on the improvements made as compared to the LLM base model.

We hope ImProver will serve as a foundation for further advancements in automated proof optimization and inspire new applications in theorem proving and ML-assisted formal reasoning.


## Revisions
-   Renamed the readability metric to declarativity to account for the subjectivity of what exactly a “readable” proof is.
-   Provided side-by-side examples of declarative versus non-declarative proofs.
-   Acknowledged the limitations of simplistic metrics in modeling complex proof structures in our initial evaluation.
-   Addressed degenerate solutions and how to mitigate them.
-   Highlighted details about input formats for ImProver and GPT-4o versioning.
-   Clarified the ablation and factorial experimentation process.
-   Added an ablation to examine the effects of syntax guidance on performance.
-   Revised the introduction to provide stronger and clearer motivation for proof optimization.
-   Expanded the discussion of related work to include interactive proofs and prover-verifier games.
-   Clarified the role of temperature in ensuring diversity for best-of-N outputs.
-   Clarified terminology to be more consistent.
-   Fixed typos.


## Ongoing Experiments
-   Extending evaluations of ImProver’s performance on additional test splits of miniF2F.
-   Adding a more complex “mixed” metric, which optimizes for both length and declarativity.


## Thank You
We once again thank the reviewers for their valuable and detailed feedback. We sincerely hope the revisions and discussions presented here demonstrate the contributions and impact of ImProver. We have responded to individual concerns in detail below and welcome any further suggestions or questions.


## References
[1] https://github.com/leanprover-community/mathlib4

[2] [https://github.com/leanprover-community/mathematics\_in\_lean](https://github.com/leanprover-community/mathematics%5C_in%5C_lean)

[3] https://teorth.github.io/pfr/

[4] https://doi.org/10.48550/arXiv.2210.12283

---

> ### Author Response · Authors · 2024-11-28
>
> Thank you again to all reviewers for your suggestions and feedback on our work. We have now submitted a new revision containing the results of the aforementioned "ongoing experiments". We look forward to your feedback and discussion on how to continue to ensure the robustness and quality of our work. We also have attached point-by-point responses to reviewer comments that are relevant to the changes made in this revision.
>
> ## Revisions
> - Implemented and tested a "mixed metric" which is a more complex metric that aims to optimize for both length and declarativity at the same time.
> - Tested ImProver's theorem proving performance on MiniF2F
> - Moved all qualitative results to appendix B.3
>
> ## Edits
> For the sake of transparency, we note that the previous comment has been edited; initially we intended to implement a dependency metric rather than a mixed metric, but after more reflection between the two, we changed our minds to implement the latter. Dependency is certainly a valuable problem to tackle, and we plan to pursue it in more detail in future works. However, for this paper, we believe that reviewer 5mDw's suggestion of a mixed metric is more relevant and useful. Additionally, our initial plans for testing theorem proving was to use the miniCTX benchmark, but we have revised to now use the miniF2F benchmark instead.

---

### Meta-Review · Area_Chair_DTPH · 2024-12-24

**Metareview:**

This paper concerns automated proof optimization and proposes ImProver, an LLM-based approach to rewriting proofs to meet user-defined metrics in Lean. The specific metrics considered in this work are length metric, readability metric, and completion metric. ImProver introduces Chain-of-states prompting, which makes internal proof state explicitly exposed to an LLM agent, and also leverages relatively standard techniques like retrieval and sampling (best of N solutions). The experimental evaluation shows that ImProver outperforms vanilla LLM-based approaches in terms of the proposed metrics. The problem of proof optimization is a new and important research problem, and there has been little research done by the neural theorem proving community previously. This paper proposes the first solution addressing neural theorem proof optimization. There are some concerns about whether the proposed metrics are meaningful for measuring readability of proofs, and whether the optimized proofs indeed have practical benefits. The contributions made in this work overweighs these concerns, which of course deserve further follow-up research works to address.

**Additional Comments On Reviewer Discussion:**

During the rebuttal period, authors clarified several confusions caused by typos and inaccurate citations. There are active discussions about the readability metric, which is tricky, subtle, and even subjective to some extent. The primary concern of using the ratio of "have" tactic is the risk of introducing redundant "have" tactics in the proof, which may further convolute or simplify / optimize proofs. To mitigate the concern, the authors suggest using a more accurate name "declarativity metric" instead. Another concern is the practical benefits of proof optimization. The argument given by the authors seems more anecdotal, and reviewer XEdk remains unconvinced due to the lack of quantitative evidence.

---

### Decision · Program_Chairs · 2025-01-22

Accept (Poster)